# Query-Specific Causal Graph Pruning Under Tiered Knowledge

**Yizuo Chen**[*]
University of California, Los Angeles
Los Angeles, CA 90095, USA
yizuo.chen@ucla.edu

**Jane E. Barker**
Amazon
Bellevue, WA 98004, USA
jbarkere@amazon.com

## Abstract

We present a systematic method for pruning edges from causal graphs by leveraging tiered knowledge. We characterize conditions under which edges can be removed from a causal graph while preserving the identifiability of (conditional) causal effects. This result enables causal identification on simplified graphs that are substantially smaller than the original graphs. The approach is particularly valuable when researchers are interested in causal relationships within specific tiers while accounting for broader influences from other tiers without fully specifying them. Building on this, we introduce a query-specific causal discovery algorithm that takes a causal query and observational data as input and returns a graph tailored specifically to that query. Through both theoretical analysis and empirical studies, we demonstrate that our discovery algorithm can achieve exponential speedups compared to the existing method when tiered knowledge is available.

## 1 Introduction

Pearl's Causal Hierarchy categorizes queries into three separate layers: associational, interventional, and counterfactual (Pearl & Mackenzie, 2018). The first layer examines statistical associations estimable from observational data, while the latter two layers involve causal interventions and require understanding causal relationships between variables typically established through experimental studies (e.g., randomized controlled trials). We focus on the second layer and consider two widely studied types of interventional queries: causal effects and conditional causal effects. A causal effect quantifies the impact of an intervention on an outcome, exemplified by the question "What is the probability that a patient will recover after the doctor instructs them to undergo surgery?", whereas a conditional causal effect further restricts the query to a subpopulation and asks "What is the probability that a patient *older than 65* will recover after the doctor instructs them to undergo surgery?"

A common approach to answering these causal queries is through causal identification. Given a causal graph $G$ and an observational distribution $\Pr(\mathbf{V})$, causal identification aims to estimate the values of these causal queries from $G$ and $\Pr(\mathbf{V})$. This gives rise to the *identifiability* problem, which checks whether the values of causal queries are *uniquely* determined by $G$ and $\Pr(\mathbf{V})$. Sound and complete methods for testing identifiability have been developed, including the Identify algorithm in (Tian & Pearl, 2003; Huang & Valtorta, 2006), the ID algorithm (Shpitser & Pearl, 2006), do-calculus (Pearl, 2009) for identifying causal effects, and the IDC algorithm (Shpitser & Pearl, 2008) for identifying conditional causal effects. Among these methods, the choice of causal graph plays a crucial role in determining both identifiability and the values of causal queries. For instance, consider the causal effect of surgery on patient recovery. The causal effect is identifiable if the graph contains a directed edge $Surgery \rightarrow Recovery$ but becomes unidentifiable if a hidden confounder introduces an additional bidirected edge $Surgery \leftrightarrow Recovery$.[1] The causal effect remains identifiable if the graph contains a reversed edge $Surgery \leftarrow Recovery$ but becomes negligible.

Recent advancements in causal inference have expanded beyond traditional frameworks by integrating background knowledge with causal graphs and data, leading to improvements in both identifi-

---

[*]This work was done during the author's internship at Amazon.

[1]The bidirected edge $A \leftrightarrow B$ means $A \leftarrow U \rightarrow B$ where $U$ is a *hidden confounder* causing both $A$ and $B$. For example, patients' symptoms are potential hidden confounders between "Surgery" and "Recovery".

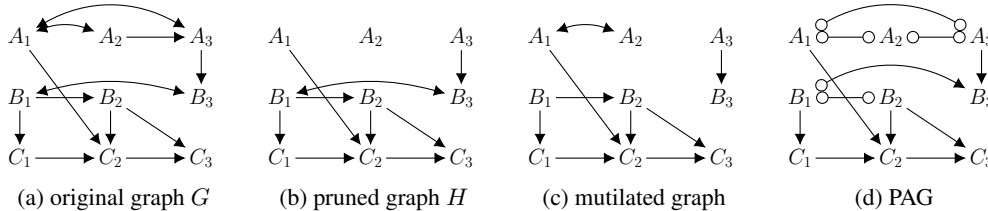

(a) original graph $G$      (b) pruned graph $H$      (c) mutilated graph      (d) PAG

Figure 1: Pruned, mutilated, and partial ancestral graph (PAG) for $G$. Tiers are indicated using distinct alphabetical letters. For example, the three tiers in the graph may represent the lecture quality, the student's learning effort, and quiz scores over three consecutive weeks.

ability and computational efficiency. These include leveraging the knowledge of functional dependencies (Darwiche, 2021; Chen & Darwiche, 2022; 2024), context-specific independencies (Tikka et al., 2019; Mokhtarian et al., 2022), and fully-known observational distributions (Chen & Darwiche, 2025). In this work, we extend this research line by showing another benefit of incorporating background knowledge: *it simplifies the causal graphs required for causal identification.* In particular, we consider tiered knowledge, which partitions variables into "tiers" with a predefined causal ordering, allowing only directed edges from higher tiers to lower tiers (Andrews et al., 2020). Such type of knowledge arises frequently, for example, in temporal reasoning where variables are ordered according to their time steps, and when exogenous context (background) variables are included in causal graphs; see (Andrews et al., 2020) for detailed discussions.

The main contribution of this work is a technique for pruning edges from causal graphs using tiered knowledge while preserving the identifiability of (conditional) causal effects. We show that this technique enables testing identifiability on simpler graphs while maintaining soundness and completeness. As a preliminary example, consider the causal graph in Figure 1a, which contains three tiers (distinguished by alphabetical letters). Our pruning method allows us to establish the identifiability of $\mathrm{Pr}_{B_1}(B_3, C_3)$ and $\mathrm{Pr}_{B_1}(B_3, C_3 | A_1)$ by showing that they are identifiable in the smaller graph $H$ (Figure 1b) constructed specifically for the given causal queries. A key advantage of using such query-specific graphs is that they reduce the effort for graph specification when edges are either determined through background knowledge or learned from data. We demonstrate this advantage by presenting a causal discovery algorithm that learns query-specific graphs from observational data, achieving exponential computational speedups over the existing method that constructs full graphs.

The paper is structured as follows. We start by reviewing some key definitions and methodologies in the literature on causal inference and discovery in Section 2. In Section 3, we present our main results on graph pruning that preserves the identifiability of both unconditional and conditional causal effects. We follow this by introducing a causal discovery algorithm for learning query-specific graphs along with an analysis of its complexity in Section 4. We provide experimental results demonstrating the effectiveness of our method in Section 5. We finally close with some concluding remarks in Section 6. All proofs are included in the Appendix.

## 2 TECHNICAL PRELIMINARIES

We assume all variables are discrete, though all the results presented in this work can be extended to continuous domains. Single variables are denoted by uppercase letters (e.g., $X$) and their states are denoted by lowercase letters (e.g., $x$). Sets of variables are denoted by bold uppercase letters (e.g., $\mathbf{X}$) and their instantiations are denoted by bold lowercase letters (e.g., $\mathbf{x}$).

### 2.1 CAUSAL GRAPHS AND TIERED KNOWLEDGE

Causal graphs are widely used to model the causal relationship among variables. In this work, we consider causal graphs in the form of Acyclic Directed Mixed Graphs (ADMGs), which contain both directed and bidirected edges and do not assume causal sufficiency.

**Definition 2.1.** (Richardson, 2003) An acyclic directed mixed graph (ADMG) is a graph that contains directed edges ($\rightarrow$) and bidirected edges ($\leftrightarrow$) and in which directed edges do not form cycles.

Figures 1a and 1b are both valid ADMGs. The edges in an ADMG are interpreted as follows. Each directed edge $A \rightarrow B$ means that $A$ is a *direct cause* of $B$; in this case, we call $A$ a *parent* of $B$ and $B$ a *child* of $A$. Each bidirected edge $A \leftrightarrow B$ indicates the existence of a *hidden (unobserved) confounder* causing both $A$ and $B$. In Figure 1a, for example, $A_1$ is a direct cause of $C_2$, and there exists a hidden confounder between $B_1$ and $B_3$. We denote the observed variables in an ADMG by $\mathbf{V}$ and the hidden confounders by $\mathbf{U}$. Variable $A$ is a *neighbor* of $B$ if there is an edge (directed or bidirected) between $A$ and $B$. Two variables $A, B$ are said to be in a same *c-component* if they are connected by a bidirected path, i.e., a path containing only bidirected edges (Tian & Pearl, 2002). The variables in a graph can always be partitioned into c-components. For example, the variables in Figure 1a can be partitioned into $\{A_1, A_2, A_3\}$, $\{B_1, B_3\}$, $\{B_2\}$, $\{C_1\}$, $\{C_2\}$, and $\{C_3\}$.

Tiered knowledge imposes additional constraints on causal graphs by specifying a causal ordering over its variables (Andrews et al., 2020). By definition, tiered knowledge partitions $\mathbf{V}$ into $t$ tiers (disjoint subsets), denoted $\{\mathbf{V}^1, \ldots, \mathbf{V}^t\}$, where only directed edges can be added from higher tiers (tiers with smaller indexes) to lower tiers (tiers with larger indexes). That is, for each pair $X \in \mathbf{V}^i$ and $Y \in \mathbf{V}^j$ where $i < j$, only the edge $X \rightarrow Y$ is permitted. Consider Figure 1a with tiers $\{A_1, A_2, A_3\}$, $\{B_1, B_2, B_3\}$, and $\{C_1, C_2, C_3\}$. The directed edge $A_1 \rightarrow C_2$ is allowed since $A_1$ belongs to a higher tier than $C_2$, whereas neither $B_2 \rightarrow A_2$ nor $B_2 \leftrightarrow A_2$ is allowed since $B_2$ belongs to a lower tier than $A_2$. All edges in Figures 1a and 1b satisfy this constraint. Such tiered knowledge commonly arises in scenarios where time-series features exhibit contemporaneous confounding, a set of context (background) variables are exogenous to the variables of interest, or features are organized into a hierarchical order; see Appendix B for further elaboration on these scenarios and (Andrews et al., 2020) for additional examples.

For simplicity, we use $\Gamma$ to denote the *tier mapping* where $\Gamma(X) = i$ for each $X \in \mathbf{V}^i$. For a set of variables $\mathbf{W}$, let $\Gamma^-(\mathbf{W})$ denote the *minimum* tier index of $\mathbf{W}$, i.e., $\Gamma^-(\mathbf{W}) = min_{W \in \mathbf{W}} \Gamma(W)$, and let $\Gamma^+(\mathbf{W})$ denote the *maximum* tier index of $\mathbf{W}$, i.e., $\Gamma^+(\mathbf{W}) = max_{W \in \mathbf{W}} \Gamma(W)$. In Figure 1a, for example, $\Gamma^-(B_1, B_3, C_2) = 2$ and $\Gamma^+(B_1, B_3, C_2) = 3$.

## 2.2 Identifying (Conditional) Causal Effects

The identification of unconditional and conditional causal effects from observational data has been studied extensively in the past; see, e.g., (Pearl, 2009; Hernán & Robins, 2020; Peters et al., 2017; Imbens & Rubin, 2015; Tian, 2004; Shpitser & Pearl, 2008). Let $G$ be a causal graph and $M$ be a model (parameterization) for $G$, an intervention $do(\mathbf{x})$ induces a mutilated graph $G_{\mathbf{X}}$ obtained from $G$ by removing all the incoming edges of $\mathbf{X}$, and a modified model $M_{\mathbf{x}}$ obtained by fixing the states of $\mathbf{X}$ to $\mathbf{x}$ in $M$. Figure 1c depicts the mutilated graph under interventions on $A_3, B_1$. The modified model $M_{\mathbf{x}}$ induces an interventional distribution, denoted $\mathrm{Pr}_{\mathbf{x}}(\mathbf{V}, \mathbf{U})$, used to compute (conditional) causal effects. Given a set of treatments $\mathbf{X}$ and a set of outcomes $\mathbf{Y}$, the *causal effect* of $\mathbf{X}$ on $\mathbf{Y}$, denoted $\mathrm{Pr}_{\mathbf{X}}(\mathbf{Y})$, computes the marginal distribution $\mathrm{Pr}_{\mathbf{x}}(\mathbf{Y})$ for each instantiation $\mathbf{x}$. The *conditional causal effect* further takes a variable set $\mathbf{Z}$ and computes $\mathrm{Pr}_{\mathbf{x}}(\mathbf{Y}|\mathbf{Z})$ for each $\mathbf{x}$.

The identifiability problem studies whether a causal query is uniquely determined by the causal graph $G$ and observational distribution $\mathrm{Pr}(\mathbf{V})$. We next review the formal definition of conditional causal effect identifiability – unconditional causal effect identifiability is a special case when $\mathbf{Z} = \emptyset$.

**Definition 2.2.** (Tian, 2004) The conditional causal effect of $\mathbf{X}$ on $\mathbf{Y}$ given $\mathbf{Z}$ is said to be identifiable with respect to $\langle G, \mathbf{V} \rangle$ if $\mathrm{Pr}^1_{\mathbf{x}}(\mathbf{y}|\mathbf{z}) = \mathrm{Pr}^2_{\mathbf{x}}(\mathbf{y}|\mathbf{z})$ for every pair of distributions $\mathrm{Pr}^1$ and $\mathrm{Pr}^2$ induced by $G$ such that $\mathrm{Pr}^1(\mathbf{V}) = \mathrm{Pr}^2(\mathbf{V}) > 0$.[2]

A causal query is called *unidentifiable* if it is not identifiable. In Figure 1a, $\mathrm{Pr}_{B_1}(B_3, C_3)$ and $\mathrm{Pr}_{A_1, B_1}(C_3|C_2)$ are both identifiable. On the other hand, $\mathrm{Pr}_{A_2}(B_3, C_2)$ and $\mathrm{Pr}_{A_2}(C_2|A_3)$ are unidentifiable. Sound and complete methods based on c-components have been developed for identifying both unconditional and conditional causal effects in the past; see, e.g., (Tian & Pearl, 2003; Huang & Valtorta, 2006; Shpitser & Pearl, 2008). Intuitively, these methods test the identifiability of (conditional) causal effects by reducing it to independent identifiability problems over the c-components in the causal graph. Moreover, they provide a formula for estimating the causal effects if they are deemed identifiable. For example, the IDENTIFY algorithm (Tian & Pearl, 2003) not

---

[2]The strict positivity here is commonly assumed by existing identification methods such as do-calculus (Pearl, 2009) and IDENTIFY (Tian & Pearl, 2003).

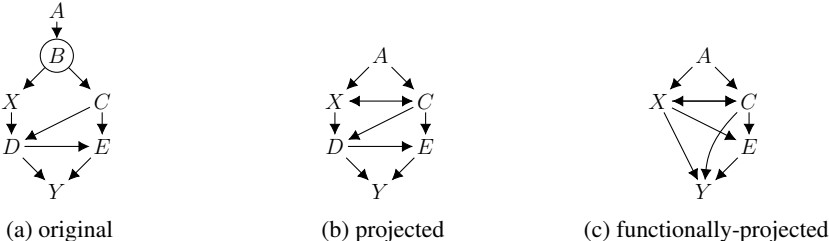

(a) original            (b) projected            (c) functionally-projected

Figure 2: Original, projected, and functionally projected graphs that preserve identifiability.

only establishes the identifiability of $\Pr_{B_1}(B_3, C_3)$ in Figure 1a but also yields the following formula: $\Pr_{b_1}(b_3, c_3) = \sum_{a_1, a_3, b_2, c_1, c_2} \Pr(a_1, a_3)\Pr(b_2|b_1)\Pr(c_1|b_1)\Pr(c_2|a_1, b_2, c_1)\Pr(c_3|b_2, c_2)$ $\sum_{b_1'} \Pr(b_3|a_3, b_1') \Pr(b_1')$; see Appendix A for additional details on the identification methods.

### 2.3 CAUSAL DISCOVERY WITH TIERED KNOWLEDGE

When causal graphs are learned from observational data, it is standard to consider the class of Maximal Ancestral Graphs (MAGs), a subclass of ADMGs that satisfy additional properties.[3] Existing causal discovery algorithms such as Fast Causal Inference (FCI) (Spirtes et al., 2000) take observational data as input and learn a Markov equivalence class (MEC) consisting of all MAGs that encode the same conditional independencies. Such MECs are commonly represented using Partial Ancestral Graphs (PAGs), which contain edges of the forms $\rightarrow$, $\leftrightarrow$, $\circ\!\!\rightarrow$ and $\circ\!\!-\!\!\circ$, where each $\circ$ can be either an arrowhead or a tail. For example, Figure 1d depicts a PAG representing the MEC that contains the MAG in Figure 1a. To utilize these PAGs for causal identification, it is common to perform a postprocessing step that recovers the true causal graph (MAG) by replacing each $\circ$ mark in the learned PAG with either an arrowheads or a tail. This can be achieved using approaches that leverage background knowledge (Meek, 1995; Heckerman et al., 1995; Borboudakis & Tsamardinos, 2012) or incorporate experimental data (He & Geng, 2008; Hauser & Bühlmann, 2014; Kocaoglu et al., 2017). Another line of work derives bounds by enumerating MAGs from the PAG and computing causal effects for each MAG (Hyttinen et al., 2015; Malinsky & Spirtes, 2017; Wang et al., 2024).[4]

When tiered knowledge is available, a modified FCI algorithm was proposed in (Andrews et al., 2020) to learn a PAG from observational data. The algorithm is sound and complete for recovering the *maximally informative* PAG that encodes the conditional independencies in the observational distribution $\Pr(\mathbf{V})$ and that respects the tiered knowledge.[5] Moreover, suppose $\Pr(\mathbf{V})$ reflects the conditional independencies implied by a causal graph $G$, the algorithm is guaranteed to return a PAG representing the Markov equivalence class that contains $G$; see Section 4 for more details.

## 3 GRAPH PRUNING UNDER TIERED KNOWLEDGE

We now present methods for pruning edges from a causal graph while preserving the identifiability of a given (conditional) causal effect under tiered knowledge. These methods allow us to reduce the identifiability problem in the original graph $G$ to the identifiability problem in a pruned graph $G'$, which is advantageous when $G'$ is easier to attain than $G$. In this case, we say that $G$ and $G'$ are *ID-equivalent* since they yield the same identifiability results for the given causal effect.

Such type of pruning-based approach for testing identifiability have been explored in prior works. A well-known example is the *(latent) projection* operation (Verma, 1993; Tian & Pearl, 2002), which removes hidden variables from a DAG before applying causal identification methods such as those

---

[3]MAGs are a subclass of ADMGs that satisfy *ancestrality* and *maximality*. Ancestrality prohibits bidirected edges between nodes $X$ and $Y$ if there is a directed path from $X$ to $Y$. Maximality requires that any non-adjacent nodes $X, Y$ can be m-separated by some other variables; see (Richardson & Spirtes, 2002) for details.

[4]An alternative approach avoids the conversion to MAGs by directly identifying causal queries from PAGs; see, e.g., (Jaber et al., 2022; Perkovic et al., 2017).

[5]Formally, a PAG is *maximally informative* if each edge mark $\circ$ in the PAG corresponds to an arrowhead in some MAG and an arrow tail in another MAG within the MEC.

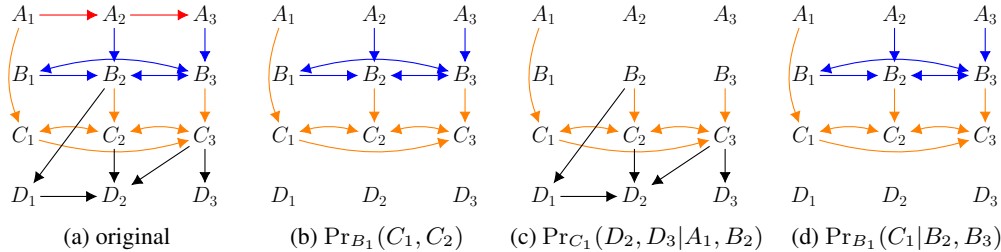

Figure 3: Original graph and its pruned graphs for identifying $\Pr_{B_1}(C_1, C_2)$, $\Pr_{C_1}(D_2, D_3|A_1, B_2)$ and $\Pr_{B_1}(C_1|B_2, B_3)$. Tiers are indicated using distinct alphabetical letters.

in (Tian & Pearl, 2003; Shpitser & Pearl, 2006). Figure 2b depicts the result of applying projection on Figure 2a when variable $B$ is hidden. Tikka & Karvanen (2017) further studied the pruning of observed variables from causal graphs to simplify the identification process. More recently, a refined projection operation called *functional projection* was introduced to eliminate (remove) variables that exhibit functional dependencies on their parents (e.g., "legal driving age" is functionally determined by "country") (Chen & Darwiche, 2024). For example, suppose variable $D$ in Figure 2b exhibits functional dependency on its parents $\{X, C\}$, then eliminating $D$ using functional projection yields the graph in Figure 2c. Both projected and functionally projected graphs are ID-equivalent to the original graph and can thus be used to test causal effect identifiability.

The pruning methods introduced in this work also construct ID-equivalent graphs but differ from the above approaches in two key aspects. First, our methods remove edges rather than variables from causal graphs. Second, our methods can be applied whenever tiered knowledge is available, whereas the previous approaches require the full causal graph to be known. The second distinction allows us to identify causal effects using a smaller graph by leveraging tiered knowledge, avoiding the need to specify (or learn) the entire causal graph as typically required in classical approaches.

## 3.1 Edge Pruning for Causal Effects

Unless stated otherwise, let $G$ denote a causal graph in which the observational variables $\mathbf{V}$ are partitioned into tiers $\{\mathbf{V}^1, \ldots, \mathbf{V}^t\}$ with tier mapping $\Gamma$. Moreover, let $\mathbf{X}$ and $\mathbf{Y}$ denote disjoint sets of treatment and outcome variables, respectively. We start by reviewing a key graphical notion introduced in (Andrews et al., 2020), which we refer to as a "T-component" for convenience.

**Definition 3.1.** (Andrews et al., 2020) The $i^{\text{th}}$ ($1 \leq i \leq t$) tiered-component (T-component) is the subgraph of $G$ containing all variables of $G$ and all edges $(l, r)$ for which $max\{\Gamma(l), \Gamma(r)\} = i$.

For simplicity, we denote the $i^{\text{th}}$ T-component as $G^i$. In Figure 3a, edges in different T-components are highlighted with distinct colors. One key result is that a causal graph can always be decomposed into T-components based on tiered knowledge, i.e., $G = \bigcup_{i=1}^t G^i$.[6]

We now present our main result which prunes edges from the original graph $G$ while preserving the identifiability of a causal effect $\Pr_{\mathbf{X}}(\mathbf{Y})$.

**Proposition 3.2.** *Let $H$ be the union of all T-components $G^i$ with $\Gamma^-(\mathbf{X}) \leq i \leq \Gamma^+(\mathbf{Y})$. Then $\Pr_{\mathbf{X}}(\mathbf{Y})$ is identifiable in $G$ iff it is identifiable in $H$ and can be computed as $\Pr_{\mathbf{x}}(\mathbf{y}) = \sum_{\mathbf{w}\backslash\mathbf{y}} \Pr(\mathbf{w})\text{IDENTIFY}_H(\mathbf{x} \cup \mathbf{w}, \mathbf{y} \backslash \mathbf{w})$, where $\mathbf{W}$ is the set of variables whose tier indexes are smaller than $\Gamma^-(\mathbf{X})$, and $\text{IDENTIFY}_H(\mathbf{x}\cup\mathbf{w}, \mathbf{y}\backslash\mathbf{w})$ denotes the formula returned by $\text{IDENTIFY}$ (Tian & Pearl, 2003) for identifying $\Pr_{\mathbf{x}\mathbf{w}}(\mathbf{y} \backslash \mathbf{w})$ in the graph $H$.[7]*

Consider the causal graph $G$ in Figure 3a and the causal effect $\Pr_{B_1}(C_1, C_2)$. Without tiered knowledge, we would typically construct the full graph $G$ containing all edges and use it to identify the causal effect. However, suppose we know in advance that $G$ can be partitioned into four tiers (indicated by different alphabetical letters), we can apply Proposition 3.2 and use the pruned graph $H$

---

[6]The union of graphs $G^1(\mathbf{V}_1, \mathbf{E}_1), \ldots, G^t(\mathbf{V}_t, \mathbf{E}_t)$ induces a new graph $G(\bigcup_{i=1}^t \mathbf{V}_i, \bigcup_{i=1}^t \mathbf{E}_i)$.

[7]The completeness of IDENTIFY was proven in (Huang & Valtorta, 2006). The IDENTIFY algorithm here can also be replaced by the ID algorithm (Shpitser & Pearl, 2006) which is also sound and complete. We cannot replace IDENTIFY with $\Pr_{\mathbf{x}\mathbf{w}}(\mathbf{y} \backslash \mathbf{w})$ here since the distribution Pr is not induced by the subgraph $H$.

shown in Figure 3b for identification. Specifically, $H$ is constructed as the union of T-components $G^2$ and $G^3$ since $\Gamma^+(\mathbf{X}) = \Gamma^+(B_1) = 2$ and $\Gamma^-(\mathbf{Y}) = \Gamma^-(C_1, C_2) = 3$. The pruned graph $H$ contains in-tier edges (e.g., $C_1 \leftrightarrow C_2$) and cross-tier edges (e.g., $A_1 \rightarrow C_1$). It is important to include cross-tier edges in pruned graphs for several reasons. First, when the pruned graph is learned from observational data using causal discovery algorithms, omitting cross-tier edges may lead to erroneous edges in the graph, which can potentially alter identifiability results. For example, ignoring $C_3 \rightarrow D_2$ and $C_3 \rightarrow D_3$ in Figure 3a would introduce a false-positive edge between $D_2$ and $D_3$. Second, cross-tier edges capture parent-child relationships that can be leveraged to simplify the identifying formulas in Proposition 3.2; see footnote 8 for a concrete example.

Proposition 3.2 establishes two key properties of the pruned graph $H$. First, the identifiability of the given causal effect is preserved, i.e., the original and pruned graphs are ID-equivalent. For example, we can apply existing methods such as IDENTIFY to establish the identifiability of $\mathrm{Pr}_{B_1}(C_1, C_2)$ in Figure 3b. Second, the proposition provides an identifying formula for estimating the causal effect from $H$ if it is identifiable. In this example, one such formula is $\mathrm{Pr}_{b_1}(c_1, c_2) = \sum_{a_1, a_2, b_2} \mathrm{Pr}(a_1, a_2)$ $\mathrm{Pr}(b_2|a_2, b_1)\mathrm{Pr}(c_2|a_1, b_2, c_1)\mathrm{Pr}(c_1|a_1)$.[8] Note that the formula in Proposition 3.2 is constructed in a particular way, as general identifying formulas derived from pruned graphs may not be valid for identifying causal effects in the original graph. To illustrate, the following formula correctly computes $\mathrm{Pr}_{B_1}(C_1, C_2)$ in $H$: $\mathrm{Pr}_{b_1}(c_1, c_2) = \sum_{b_2}[\sum_{a_1} \mathrm{Pr}(c_2|a_1, b_2, c_1) \, \mathrm{Pr}(a_1, c_1)] \, [\sum_{a_2} \mathrm{Pr}(a_2)$ $\mathrm{Pr}(b_2|a_2, b_1)]$, yet it does not compute the causal effect in the original graph in Figure 3a and hence cannot be used. Such errors occur since the pruned graphs ignore dependencies among variables in tiers above treatment variables $\mathbf{X}$, which are essential for estimating causal effects correctly.

A remaining question is whether the ID-equivalent causal graphs constructed by Proposition 3.2 are optimal in size. Finding *minimal* ID-equivalent causal graphs offers significant benefits for causal graph specification. For graphs constructed from human knowledge, only edges in the minimal graphs need to labeled, which can save significant manual effort when the original graphs are large or when some edge orientations are unknown. For graphs learned from data, using smaller graphs can substantially reduce the computational cost of causal discovery as we demonstrate later.

We show that the bounds $\Gamma^-(\mathbf{X})$ and $\Gamma^+(\mathbf{Y})$ provided in Proposition 3.2 are tight. Specifically, the identifiability of causal effects may no longer be preserved if we prune additional edges from $G$ by increasing $\Gamma^-(\mathbf{X})$ or decreasing $\Gamma^+(\mathbf{Y})$. This holds trivially when $\Gamma^-(\mathbf{X}) > \Gamma^+(\mathbf{Y})$ since the pruned graph contains no edges. The next proposition proves the tightness for all remaining cases.

**Proposition 3.3.** *Let $\mathcal{L}, \mathcal{U}, \mathcal{L}', \mathcal{U}'$ be positive integers satisfying $\mathcal{L} \le \mathcal{U}$, $\mathcal{L}' \le \mathcal{U}'$, and at least one of $\mathcal{L}' > \mathcal{L}$ or $\mathcal{U}' < \mathcal{U}$ holds. Then there exists a causal graph $G$ and a tier mapping $\Gamma$ for which $\Gamma^-(\mathbf{X}) = \mathcal{L}$, $\Gamma^+(\mathbf{Y}) = \mathcal{U}$, and Proposition 3.2 no longer holds if the bounds $\Gamma^-(\mathbf{X})$ and $\Gamma^+(\mathbf{Y})$ are replaced with $\mathcal{L}'$ and $\mathcal{U}'$.*

Before discussing edge pruning for conditional causal effects, we highlight an important result on the size of pruned graphs: there exists a class of graphs where the sizes of original graphs are unbounded, while the sizes of their pruned graphs (from Proposition 3.2) are bounded. Such graphs can be constructed, for example, by adding more tiers to Figure 3a. As more tiers are added above $B_i$'s, the size of the original graph grows to infinity, whereas the size of the pruned graph used to identify $\mathrm{Pr}_{B_1}(C_1, C_2)$ remains constant. This result has important implications for graph specification: when causal graphs are constructed from background knowledge or learned via causal discovery, it may suffice to specify or learn a constant number of edges to identify the causal effect, even if the full causal graph is arbitrarily large. We formalize this result in the following proposition.

**Proposition 3.4.** *There exists a class of causal graphs $G^t$ with $t$ tiers such that the number of edges in $G^t$ grows without bound as $t$ increases, while the corresponding pruned graphs (Proposition 3.2) contain a constant number of edges.*

## 3.2 EDGE PRUNING FOR CONDITIONAL CAUSAL EFFECTS

We now consider another class of causal queries called conditional causal effects. These queries generalize causal effects by taking an additional set of conditioning variables $\mathbf{Z}$ as input and compute $\mathrm{Pr}_{\mathbf{X}}(\mathbf{Y}|\mathbf{Z})$. Analogous to the case of causal effects, we introduce a method that prunes edges from

---

[8]The formula is simplified from the output of IDENTIFY: $\mathrm{Pr}_{b_1}(c_1, c_2) = \sum_{a_1, a_2, a_3, b_2} \mathrm{Pr}(a_1, a_2, a_3)$ $\mathrm{Pr}(b_2|a_2, b_1) \, \mathrm{Pr}(c_2|a_1, b_2, c_1) \, \mathrm{Pr}(c_1|a_1)$. This illustrates the benefit of including cross-tier edges.

causal graphs while preserving the identifiability of $\Pr_{\mathbf{X}}(\mathbf{Y}|\mathbf{Z})$. The pruning criterion, however, is more subtle for conditional causal effects as it depends on the choice of conditioning variables $\mathbf{Z}$.

We start with the case where variables in $\mathbf{Z}$ belong to higher tiers than the treatments $\mathbf{X}$.

**Proposition 3.5.** *Let $H$ be the union of all T-components $G^i$ with $\Gamma^-(\mathbf{X}) \leq i \leq \Gamma^+(\mathbf{Y})$. If $\Gamma^+(\mathbf{Z}) < \Gamma^-(\mathbf{X})$, then $\Pr_{\mathbf{X}}(\mathbf{Y}|\mathbf{Z})$ is identifiable in $G$ iff it is identifiable in $H$ and can be computed as $\Pr_{\mathbf{x}}(\mathbf{y}|\mathbf{z}) = \frac{\Pr_{\mathbf{x}}(\mathbf{y},\mathbf{z})}{\Pr(\mathbf{z})}$.*

That is, a conditional causal effect is identifiable in the original graph $G$ iff it is identifiable in the pruned graph $H$. We stress again that this reduction only holds when the conditioning variables $\mathbf{Z}$ are in tiers above the treatment variables $\mathbf{X}$. The identifying formula contains two parts: an interventional probability $\Pr_{\mathbf{x}}(\mathbf{y},\mathbf{z})$ and an observational probability $\Pr(\mathbf{z})$. While $\Pr(\mathbf{z})$ can be directly computed from the observational distribution, Proposition 3.2 can be applied to estimate $\Pr_{\mathbf{x}}(\mathbf{y},\mathbf{z})$ by setting the treatments to $\mathbf{X}' = \mathbf{X}$ and the outcomes to $\mathbf{Y}' = \mathbf{Y} \cup \mathbf{Z}$. Interestingly, the pruned graph constructed for $\Pr_{\mathbf{X}'}(\mathbf{Y}')$ using Proposition 3.2 is always identical to the pruned graph constructed for $\Pr_{\mathbf{X}}(\mathbf{Y})$ using Proposition 3.5 when $\Gamma^+(\mathbf{Z}) < \Gamma^-(\mathbf{X})$.

Consider again the causal graph in Figure 3a but with a conditional causal effect $\Pr_{B_1}(C_1, C_2|A_2)$. Since $\Gamma^+(\mathbf{Z}) = \Gamma^+(A_2) = 1$ is less than $\Gamma^-(\mathbf{X}) = \Gamma^-(B_1) = 2$, we can apply Proposition 3.5 and test identifiability in the pruned graph $H$ in Figure 3b. Specifically, the conditional causal effect is identifiable in $H$ according to the IDC algorithm (Shpitser & Pearl, 2006) and can be computed as $\Pr_{b_1}(c_1, c_2|a_2) = \frac{\Pr_{b_1}(c_1, c_2, a_2)}{\Pr(a_2)}$, where $\Pr_{b_1}(c_1, c_2, a_2)$ is further identified from $H$ as $\Pr_{b_1}(c_1, c_2, a_2) = \sum_{a_1, b_2} \Pr(a_1, a_2) \Pr(b_2|a_2, b_1) \Pr(c_2|a_1, b_2, c_1) \Pr(c_1|a_1)$ using Proposition 3.2. Our second example examines the same graph but a different conditional causal effect $\Pr_{C_1}(D_2, D_3|A_1, B_2)$. Since $\Gamma^+(A_1, B_2) < \Gamma^-(C_1)$, we can again apply Proposition 3.5 and test identifiability using the pruned graph in Figure 3c, which concludes that $\Pr_{C_1}(D_2, D_3|A_1, B_2)$ is unidentifiable by the IDC algorithm. Similar to the case of unconditional causal effects, the bounds in Proposition 3.6 are tight, which we formulate and prove as Proposition D.2 in the Appendix.

One may naturally ask whether the edge pruning method remains valid when the condition $\Gamma^+(\mathbf{Z}) < \Gamma^-(\mathbf{X})$ is violated. The answer is no. To illustrate, consider the causal graph in Figure 3a and the conditional causal effect $\Pr_{B_1}(C_1|B_2, B_3)$. The condition required by Proposition 3.5 no longer holds since $\Gamma^+(B_2, B_3) = \Gamma^-(B_1)$. In this case, the causal effect is unidentifiable in the original graph according to the IDC algorithm but becomes identifiable in the pruned graph shown in Figure 3d. This example demonstrates that the edge pruning method does not, in general, preserve the identifiability of conditional causal effects when a conditioning variable lies in the same or a lower tier than a treatment variable under tiered knowledge.

However, we can always prune edges in tiers below $(\mathbf{Y} \cup \mathbf{Z})$ while preserving the identifiability of $\Pr_{\mathbf{X}}(\mathbf{Y}|\mathbf{Z})$. We state this result as Proposition 3.6 and illustrate with an example below.

**Proposition 3.6.** *Let $H$ be the union of all T-components $G^i$ with $i \leq \Gamma^+(\mathbf{Y} \cup \mathbf{Z})$. Then $\Pr_{\mathbf{X}}(\mathbf{Y}|\mathbf{Z})$ is identifiable in $G$ iff it is identifiable in $H$. Moreover, any formula that computes $\Pr_{\mathbf{X}}(\mathbf{Y}|\mathbf{Z})$ in $H$ also correctly computes it in $G$.*

Consider again the graph in Figure 3a and the conditional causal effect $\Pr_{B_1}(C_1|B_2, B_3)$. The figure on the right depicts the graph resulting from Proposition 3.6. In contrast to Figure 3d, we prune all edges below the $C_i$'s while retaining all edges at or above them. We can then apply the IDC algorithm to the pruned graph and conclude that $\Pr_{B_1}(C_1|B_2, B_3)$ is unidentifiable. Although the edge pruning described by Proposition 3.6 appears to be more conservative than that in Proposition 3.5, it is applicable regardless of the choice of conditioning variables and can still improve the efficiency of existing causal discovery algorithms, which we demonstrate in the following sections.

# 4  QUERY-SPECIFIC CAUSAL DISCOVERY

We now present a key application of edge pruning methods introduced in the previous sections: accelerating causal discovery when causal graphs are learned from data. This task happens frequently in practice when causal graphs are not available from background knowledge and hence need to be

---

**Algorithm 1** Query-Specific Causal Discovery Under Tiered Knowledge

---

**Input:** Variables $\mathbf{V} = \{\mathbf{V}^1, \ldots, \mathbf{V}^t\}$ with $t$ tiers; conditional causal effect $\mathrm{Pr}_{\mathbf{X}}(\mathbf{Y}|\mathbf{Z})$
**Output:** PAG $\mathcal{P}$

1: /* Preprocessing to determine the min and max tier indexes */
2: **if** $\mathbf{Z} = \emptyset$ or $\Gamma^+(\mathbf{Z}) < \Gamma^-(\mathbf{X})$ **then**
3:     $maxTier \leftarrow \Gamma^+(\mathbf{Y})$, $minTier \leftarrow \Gamma^-(\mathbf{X})$
4: **else**
5:     $maxTier \leftarrow \Gamma^+(\mathbf{Y} \cup \mathbf{Z})$, $minTier \leftarrow 1$
6: **end if**
7: /* The following is adapted from (Andrews et al., 2020, Algorithm 1) */
8: $G \leftarrow$ unconnected graph over $\mathbf{V}$
9: **for** $i = maxTier$ to $minTier$ **do**
10:     $\mathbf{A}_i \leftarrow \bigcup_{j=1}^{i-1} \mathbf{V}^j$
11:     $\mathbf{B}_i \leftarrow \mathbf{V}^i$
12:     $\mathcal{P}^i \leftarrow \textsc{FciExogenous}(\mathbf{A}_i, \mathbf{B}_i)$             ▷ (Andrews et al., 2020, Algorithm 2)
13:     $\mathcal{P} \leftarrow \mathcal{P} \cup \mathcal{P}^i$
14: **end for**

---

inferred from data. From now on, we assume that the causal graphs are in form of MAGs,[9] which is commonly assumed in existing causal discovery algorithms that account for hidden confounders; see, e.g., the setups of FCI (Spirtes et al., 2000) and GFCI (Ogarrio et al., 2016). When the goal is to compute (or bound) a causal effect, it is standard to first apply causal discovery algorithms to learn a Markov equivalence class (MEC) represented by a PAG, and then extract a MAG (or multiple MAGs) from the class to estimate the causal effect; see Section 2.3 for our earlier discussion. Once a MAG is obtained, we can apply the methods in Propositions 3.2-3.6 to identify the causal effects.[10]

We show how the pruning methods introduced in earlier sections can be applied to speed up causal discovery when tiered knowledge is available. In particular, we extend these methods to PAGs so that it suffices to learn smaller PAGs tailored to the causal query of interest, which reduces overall computational cost. Given a *full* PAG whose variables are partitioned into tiers, the corresponding *partial* PAG can be obtained by applying Propositions 3.2-3.6, replacing causal graphs with PAGs. For example, to identify the causal effect $\mathrm{Pr}_{B_1}(C_2)$ in Figure 4a, we only need to learn the partial PAG in Figure 4d from observational data, rather than the full PAG in Figure 4b. This is because the pruning methods in Propositions 3.2-3.6 enable identification using the smaller MAG shown in Figure 4c, whose Markov equivalence class is captured exactly by the partial PAG.

Algorithm 1 presents the detailed procedure for query-specific causal discovery. The algorithm follows a structure similar to the FCITIERS algorithm introduced in (Andrews et al., 2020) but includes an additional preprocessing step in lines 1-6. Given a conditional causal effect $\mathrm{Pr}_{\mathbf{X}}(\mathbf{Y}|\mathbf{Z})$, the preprocessing step identifies the remaining T-components in the partial PAG after pruning edges according to Proposition 3.2-3.6. For example, $G^2$ and $G^3$ are the only remaining T-components after pruning edges from the PAG in Figure 4b for the causal effect $\mathrm{Pr}_{B_1}(C_2)$. The algorithm then iterates over each remaining T-component, learning its edges using the FCIEXOGENOUS procedure described in (Andrews et al., 2020) (lines 7-14); see Appendix C for further details. In this example, learning T-components $G^2$ and $G^3$ yields the partial PAG in Figure 4d.

The following result states that Algorithm 1 is sound and complete for learning partial PAGs. That is, the algorithm is guaranteed to find the maximally informative PAG when the observational distribution reflects all the conditional independencies implied by the true causal graph.

**Proposition 4.1.** *Let $G$ be a MAG, $\mathcal{P}$ be the maximally informative PAG for the MEC of $G$ under tiered knowledge, and $\mathcal{P}'$ be the result of pruning edges from $\mathcal{P}$ according to Propositions 3.2-3.6. Then Algorithm 1 returns $\mathcal{P}'$ when using $G$ as the conditional independence oracle.*

---

[9]Results in previous sections hold for MAGs as well since they are a subclass of ADMGs.

[10]To illustrate, the edge $C_3 \circ\!\!\rightarrow C_2$ in the PAG in Figure 4d may be oriented as $C_3 \leftrightarrow C_2$ based on background knowledge, yielding a MAG in Figure 4c which is used to identify the causal effect $\mathrm{Pr}_{B_1}(C_2)$. Causal identification methods (e.g., IDENTIFY, ID, IDC) run in polynomial time and are much faster than causal discovery, so the time required for identifiability checks is negligible here.

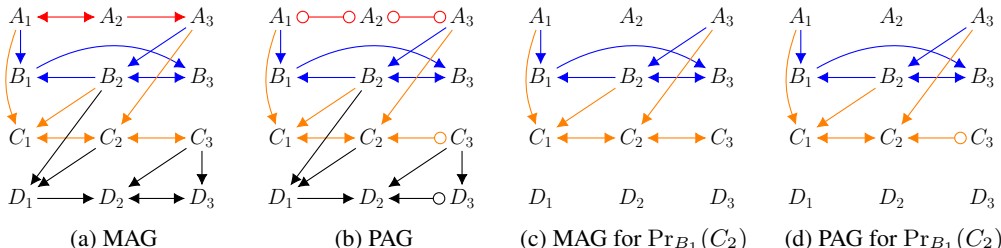

Figure 4: MAG $G$, PAG for $G$, MAG required for $\mathrm{Pr}_{B_1}(C_2)$, and PAG learned from Algorithm 1.

A key benefit of focusing on a partial (smaller) PAG is the reduction in computational cost during causal discovery. In Section 2.3, we presented a class of causal graphs whose size grows unboundedly with an increasing number of tiers, but which can be reduced to a bounded size with edge pruning. This example remains relevant for causal discovery, where learning full PAGs can be computationally unbounded, while learning the corresponding partial PAGs is tractable. Here, we show an additional result demonstrating that the edge pruning technique can yield exponential speedups in causal discovery even when the number of tiers is fixed. This happens when most edges are located in the higher tiers of PAGs and can therefore be pruned, avoiding the need to perform an exponential number of conditional independence tests to learn these edges.

**Proposition 4.2.** *There exists a class of causal effects and distributions induced by causal graphs with $n$ nodes and 2 tiers for which* FCITIERS *(Andrews et al., 2020) takes $O(n^2 \cdot 2^n)$ time, whereas Algorithm 1 takes $O(n^4)$ time, assuming access to a conditional independence oracle.*

## 5 EXPERIMENTS

We present empirical results to demonstrate the improved computational efficiency enabled by our query-specific causal discovery algorithm. Specifically, we compare the execution times of FCITIERS (Andrews et al., 2020), which learns full PAGs, and our Algorithm 1, which learns partial PAGs, on randomly generated causal graphs and (conditional) causal effects.

We start by generating random causal graphs with $T \in \{5, 6, 7, 8, 9, 10\}$ tiers, where each tier contains 10 variables that have either two or three states. We randomly assign at most 5 parents for each variable and randomly convert 30% of the in-tier (directed) edges into bidirected edges. To ensure sparsity, we bound the number of neighbors for each node by 4. We then randomly assign a parameterization for the causal graph and sample $500,000$ data instances from the model. Finally, we generate a conditional causal effect by randomly selecting 3 treatment variables, 3 outcome variables, and a conditioning set of size $\{0, 1, 2, 3\}$ from the causal graph.

Figure 5 compares the execution times of FCITIERS (Andrews et al., 2020) and Algorithm 1 for learning PAGs from generated samples, averaged over 30 runs. We observe the following patterns. First, Algorithm 1 achieves shorter execution times in all cases, which matches our expectation that edge pruning accelerates the learning of partial PAGs. Moreover, for a fixed number of conditioning variables, the performance gap between the two algorithms widens as the number of tiers in the causal graph increases, since additional tiers provide more opportunities for edge pruning. Second, for a fixed causal graph, increasing the size of the conditioning set $\mathbf{Z}$ reduces the performance gap. This happens because the constraint $\Gamma^+(\mathbf{Z}) < \Gamma^-(\mathbf{X})$ in Proposition 3.5 is less likely to hold under a larger $\mathbf{Z}$, resulting in less edge pruning.

In addition to computation time, we evaluate the precision and recall of the adjacencies and arrowheads learned by both methods, as reported in Table 1 in the Appendix.[11] In particular, the PAGs returned by each method are compared against the ground truth obtained by running FCITIERS with the true causal graph as the conditional independence oracle. The results show that Algorithm 1

---

[11]The adjacency (arrowhead) precision and recall are used in (Scheines & Ramsey, 2016) to evaluate the soundness and completeness of causal discovery algorithms. Adjacency (arrowhead) precision measures the proportion of predicted edges (arrowheads) that are present in the true PAG, and adjacency (arrowhead) recall measures the proportion of true edges (arrowheads) correctly predicted by the algorithm.

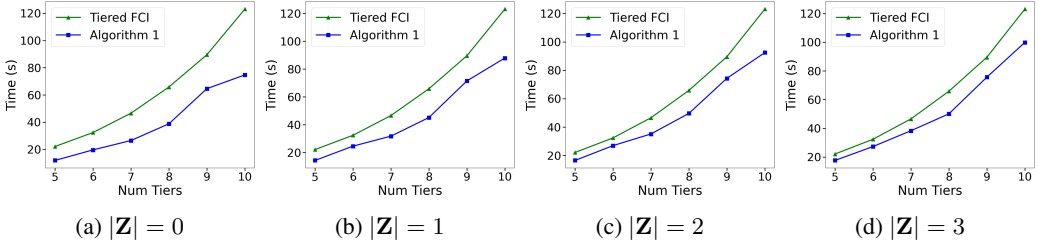

(a) $|\mathbf{Z}| = 0$      (b) $|\mathbf{Z}| = 1$      (c) $|\mathbf{Z}| = 2$      (d) $|\mathbf{Z}| = 3$

Figure 5: Comparison of execution times between tiered FCI (Andrews et al., 2020) and Algorithm 1 for varying sizes of the conditioning set $\mathbf{Z}$. The queries are reduced to causal effects when $|\mathbf{Z}| = 0$.

achieves precisions and recalls comparable to FCITIERS, supporting the soundness and completeness of Algorithm 1 in identifying partial PAGs.

## 6    CONCLUSION

We proposed a graph pruning method that removes edges from causal graphs while preserving the identifiability of (conditional) causal effects under tiered knowledge. This approach can enable more efficient specification of causal graphs for causal effect identification, whether the graphs are constructed from background knowledge or learned from observational data. In this work, we focused on the latter case and developed a causal discovery algorithm that incorporates graph pruning as a preprocessing step, achieving significant speedups over the existing method. Potential future research directions include extending the pruning methods to additional query types such as counterfactuals and generalizing the framework to handle bidirected edges across tiers.

## ACKNOWLEDGMENTS

This work is supported by Amazon. The authors would like to thank Shreya Sarawgi, Mauro Florez, Heather Hodges, and Megan Kinney for all the useful discussions and feedback.

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

# A METHODS FOR IDENTIFYING (CONDITIONAL) CAUSAL EFFECTS

We review the IDENTIFY algorithm (Tian & Pearl, 2003) for unconditional causal effects $\Pr_\mathbf{X}(\mathbf{Y})$ and the IDC algorithm (Shpitser & Pearl, 2008) for conditional causal effects $\Pr_\mathbf{X}(\mathbf{Y}|\mathbf{Z})$. Both algorithms are sound and complete for testing identifiability and return a formula for estimating the (conditional) causal effects if they are identifiable.

The IDENTIFY algorithm first constructs a mutilated graph $An(\mathbf{Y})_{G_\mathbf{X}}$ by removing treatments $\mathbf{X}$ from the original graph and retaining variables that are ancestors of outcomes $\mathbf{Y}$. It then decomposes the variables in the mutilated graph into c-components $\mathcal{S} = \{\mathbf{S}_1, \ldots, \mathbf{S}_k\}$ and reduces the identifiability of $\Pr_\mathbf{X}(\mathbf{Y})$ to the identifiability of c-components. In particular, $\Pr_\mathbf{X}(\mathbf{Y})$ is identifiable iff identifiability can be established for each $\mathbf{S}_i \in \mathcal{S}$ (Huang & Valtorta, 2006).

For each c-component $\mathbf{S}_i$, there exists a c-component $\mathbf{C}$ in the original graph $G$ such that $\mathbf{C} \supseteq \mathbf{S}_i$. Let $T$ be the subgraph formed by variables in $\mathbf{S}_i$ and $H$ be the subgraph formed by variables in $\mathbf{C}$. The c-component $\mathbf{S}_i$ is deemed identifiable iff $T$ can be obtained from $H$ by repeatedly applying the following operations: (1) replace $H$ with the subgraph formed by the c-component containing $\mathbf{S}_i$, or (2) removing from $H$ all variables that are not ancestors of $\mathbf{S}_i$; see (Tian & Pearl, 2003) for the detailed algorithm. This procedure for testing identifiability has been shown to be both sound and complete in (Huang & Valtorta, 2006) and (Shpitser & Pearl, 2006).

The IDC algorithm (Shpitser & Pearl, 2008) tests the identifiability of conditional causal effects $\Pr_\mathbf{X}(\mathbf{Y}|\mathbf{Z})$ by reducing the problem to the identifiability of unconditional causal effects. The reduction involves finding a *maximal* set of variables $\mathbf{W}$ such that $\Pr_\mathbf{X}(\mathbf{Y}|\mathbf{Z}) = \Pr_{\mathbf{X}\mathbf{W}}(\mathbf{Y}|\mathbf{Z} \setminus \mathbf{W})$, using rule 2 of do-calculus (Pearl, 2009). Existing methods for identifying causal effects — such as IDENTIFY and ID (Shpitser & Pearl, 2006) — can then be applied to test the identifiability of $\Pr_{\mathbf{X}\mathbf{W}}(\mathbf{Y}|\mathbf{Z} \setminus \mathbf{W})$; see (Shpitser & Pearl, 2008) for further details.

# B REAL-WORLD SCENARIOS OF TIERED KNOWLEDGE

We present several real-world scenarios where tiered knowledge can occur. The first two were discussed in (Andrews et al., 2020), while the third is introduced in this paper.

The first scenario arises when the causal graph models time-series features with contemporaneous confounding (Andrews et al., 2020); that is, the hidden confounders only influence variables within the same timestamp but not across different timestamps. For example, consider a classroom setting where the lecture quality, the student's learning effort, and the quiz scores are recorded every week and hence can be divided into tiers by their timestamp. While directed edges may exist across weeks (e.g., a student's quiz score in one week can affect their learning effort in the next week), hidden confounders — such as holidays that simultaneously impact both lecture quality and student's effort — only affect variables within a same week since their effects rarely persist across weeks.

In the second scenario, features are divided into context (background) variables and system variables (Andrews et al., 2020). In particular, the context variables are *exogenous* to the system variables and hence no hidden confounder exists between these two sets of variables. Consider a medical example in which the context variables describe the month of the year ($M$), the region ($R$), and the temperature ($T$), whereas the system variables represent whether individuals become sick ($S$) and whether they recover ($Y$). The figure below depicts a causal graph over these variables. Directed edges may exist among context variables or from context variables to system variables. For example, month and region determine the temperature, and temperature affects the likelihood of people getting sick, as low temperatures facilitate the spread of influenza by increasing indoor crowding and enhancing 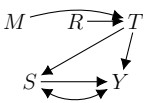 virus stability in cold, dry air. Moreover, hidden confounder may exist between system variables; for example, sickness and recovery may be confounded by unobserved variables (e.g., immunity). However, there is no plausible hidden confounder between context variables and system variables.

The third scenario occurs when variables can be grouped into different tiers within a hierarchy, with confounding restricted to occur only within each tier. To illustrate, consider an education system consisting of three tiers corresponding to the district, school, and student levels. The district level contains variables such as total district education funding and demographic features. The school level contains variables such as teaching quality and extracurricular activities. The student level

contains variables such as test grades and college admission outcomes. Hidden confounders (e.g., parental involvement and student motivation) may exist among variables within each level but do not span across different levels in general.

## C CAUSAL DISCOVERY WITH TIERED KNOWLEDGE

We review the FCIEXOGENOUS procedure introduced in (Andrews et al., 2020), which is used in our query-specific causal discovery (Algorithm 1).

Specifically, the FCIEXOGENOUS procedure recovers the edges in the $i^{\text{th}}$ T-component from observational data. It takes two sets of variables $\mathbf{A}$ and $\mathbf{B}$ as input, where $\mathbf{B}$ contains the variables in tier $i$ and $\mathbf{A}$ contains the variables above tier $i$. The procedure learns the T-component by following the standard FCI algorithm (Spirtes et al., 2000) but with two restrictions. First, edges among variables in $\mathbf{A}$ are forbidden. Second, every edge between $A \in \mathbf{A}$ and $B \in \mathbf{B}$ is oriented as $A \to B$ before applying orientation rules $\mathcal{R}_0 - \mathcal{R}_{10}$ in (Zhang, 2008); see Algorithm 2 for the detailed procedure.

## D PROOFS

### D.1 PROOF OF PROPOSITION 3.2

We start with the following lemma that shows an identifying formula for $\mathrm{Pr}_\mathbf{X}(\mathbf{Y})$ in $G$.

**Lemma D.1.** *The causal effect $\mathrm{Pr}_\mathbf{X}(\mathbf{Y})$ is identifiable in $G$ and can be computed as $\mathrm{Pr}_\mathbf{x}(\mathbf{y}) = \sum_{\mathbf{w}\setminus\mathbf{y}} \mathrm{Pr}(\mathbf{w}) \mathrm{Pr}_{\mathbf{xw}}(\mathbf{y} \setminus \mathbf{w})$ iff $\mathrm{Pr}_{\mathbf{xw}}(\mathbf{y} \setminus \mathbf{w})$ is identifiable in $G$, where $\mathbf{W}$ denotes variables whose tier indexes are smaller than $\Gamma^-(\mathbf{X})$.*

*Proof.* We first prove that $\mathrm{Pr}_\mathbf{x}(\mathbf{y}) = \sum_{\mathbf{w}\setminus\mathbf{y}} \mathrm{Pr}(\mathbf{w}) \mathrm{Pr}_{\mathbf{xw}}(\mathbf{y} \setminus \mathbf{w})$ in all graphs that satisfy tiered knowledge using do-calculus (Pearl, 2009).

$$
\begin{aligned}
\mathrm{Pr}_\mathbf{x}(\mathbf{y}) &= \sum_{\mathbf{w}\setminus\mathbf{y}} \mathrm{Pr}_\mathbf{x}(\mathbf{y}, \mathbf{w}) = \sum_{\mathbf{w}\setminus\mathbf{y}} \mathrm{Pr}_\mathbf{x}(\mathbf{w}) \mathrm{Pr}_\mathbf{x}((\mathbf{y} \setminus \mathbf{w})|\mathbf{w}) \\
&= \sum_{\mathbf{w}\setminus\mathbf{y}} \mathrm{Pr}(\mathbf{w}) \mathrm{Pr}_\mathbf{x}((\mathbf{y} \setminus \mathbf{w})|\mathbf{w}) \quad \text{(do-calculus, Rule 3 and } \Gamma^+(\mathbf{W}) < \Gamma^-(\mathbf{X})) \\
&= \sum_{\mathbf{w}\setminus\mathbf{y}} \mathrm{Pr}(\mathbf{w}) \mathrm{Pr}_{\mathbf{xw}}(\mathbf{y} \setminus \mathbf{w}) \quad \text{(do-calculus, Rule 2 and } \Gamma^+(\mathbf{W}) < \Gamma^-(\mathbf{Y} \setminus \mathbf{W}))
\end{aligned}
\tag{1}
$$

This immediately implies that $\mathrm{Pr}_\mathbf{X}(\mathbf{Y})$ is identifiable and can be computed using the formula if $\mathrm{Pr}_{\mathbf{XW}}(\mathbf{Y} \setminus \mathbf{W})$ is identifiable. We are left to show that $\mathrm{Pr}_{\mathbf{XW}}(\mathbf{Y} \setminus \mathbf{W})$ is identifiable if $\mathrm{Pr}_\mathbf{X}(\mathbf{Y})$ is identifiable. Suppose $\mathrm{Pr}_{\mathbf{XW}}(\mathbf{Y} \setminus \mathbf{W})$ is unidentifiable, there exists a hedge $\langle H, H' \rangle$ for $\mathrm{Pr}_{\mathbf{XW}}(\mathbf{Y} \setminus \mathbf{W})$ where $H' \subset H$ are c-forests in $G$ that share a same set of roots (Shpitser & Pearl, 2006).[12] Moreover, $H'$ must only contain variables that are ancestors of $(\mathbf{Y} \setminus \mathbf{W})$ in the mutilated graph $G_{\mathbf{XW}}$ in which all incoming edges of $\mathbf{X}$ and $\mathbf{W}$ are removed. This implies that all variables in $H'$ have a tier index between $\Gamma^-(\mathbf{X})$ and $\Gamma^+(\mathbf{Y} \setminus \mathbf{W})$ (no hedge exists if $\Gamma^+(\mathbf{Y} \setminus \mathbf{W}) < \Gamma^-(\mathbf{X})$). Since $H$ is a c-component containing $H'$, it also contains variables whose tier indexes are between $\Gamma^-(\mathbf{X})$ and $\Gamma^+(\mathbf{Y} \setminus \mathbf{W})$ since no bidirected edge is allowed across tiers. In this case, $\langle H, H' \rangle$ is also a hedge for $\mathrm{Pr}_\mathbf{X}(\mathbf{Y})$, which concludes the unidentifiability of $\mathrm{Pr}_\mathbf{X}(\mathbf{Y})$. $\square$

*Proof of Proposition 3.2.* By Lemma D.1, $\mathrm{Pr}_\mathbf{X}(\mathbf{Y})$ is identifiable in $G$ iff $\mathrm{Pr}_{\mathbf{XW}}(\mathbf{Y} \setminus \mathbf{W})$ is identifiable in $G$. Similarly, $\mathrm{Pr}_\mathbf{X}(\mathbf{Y})$ is identifiable in $H$ iff $\mathrm{Pr}_{\mathbf{XW}}(\mathbf{Y} \setminus \mathbf{W})$ is identifiable in $H$. Hence, it suffices to show (1) $\mathrm{Pr}_{\mathbf{XW}}(\mathbf{Y} \setminus \mathbf{W})$ is identifiable in $G$ iff it is identifiable in $H$, and (2) the formula returned by IDENTIFY$(\mathbf{x} \cup \mathbf{w}, \mathbf{y} \setminus \mathbf{w})$ in $H$ is valid for identifying $\mathrm{Pr}_{\mathbf{xw}}(\mathbf{y} \setminus \mathbf{w})$ in $G$.

First observe that $An(\mathbf{Y} \setminus \mathbf{W})_{G_{\mathbf{XW}}}$ and $An(\mathbf{Y} \setminus \mathbf{W})_{H_{\mathbf{XW}}}$ are identical, where $An(\mathbf{V})_G$ denotes the ancestors of variables $\mathbf{V}$ in graph $G$. Hence, we obtain the same c-component decomposition

---

[12] A *c-forest* is a c-component in which all nodes have at most one child. The *roots* of a c-component are the nodes that do not have any child.

---

**Algorithm 2** FCIEXOGENOUS (Andrews et al., 2020, Algorithm 2)

---

**Input:** Variable sets $\mathbf{A}, \mathbf{B}$
**Output:** PAG $\mathcal{P}$

1: Add $X_i \rightarrow X_j$ to $\mathcal{P}$ for all $X_i \in \mathbf{A}$ and $X_j \in \mathbf{B}$
2: Add $X_i \circ\!\!-\!\!\circ X_j$ to $\mathcal{P}$ for all distinct $X_i, X_j \in \mathbf{B}$
3: $n \leftarrow 0$
4: **repeat**
5:     **for all** adjacent $(X_i, X_j) \in \mathbf{A} \cup \mathbf{B}$ and subset $\mathbf{S} \subseteq adj(X_i) \setminus \{X_j\}$ where $|\mathbf{S}| = n$ **do**
6:         **if** $X_i \perp\!\!\!\perp X_j | \mathbf{S}$ **then**
7:             Delete edge $(X_i, X_j)$ from $\mathcal{P}$
8:             $sepset(X_i, X_j) \leftarrow \mathbf{S}, sepset(X_j, X_i) \leftarrow \mathbf{S}$
9:         **end if**
10:     **end for**
11:     $n \leftarrow n + 1$
12: **until** $n > |adj(X_i) \setminus \{X_j\}|$ for all adjacent $(X_i, X_j) \in \mathbf{A} \cup \mathbf{B}$
13: Apply $\mathcal{R}_0$ to $\mathcal{P}$
14: **for all** adjacent $(X_i, X_j) \in \mathbf{A} \cup \mathbf{B}$ **do**
15:     */* Checking possible d-separating sets pds */*
16:     **if** there exists $\mathbf{S} \in pds(X_i, X_j)$ where $X_i \perp\!\!\!\perp X_j | \mathbf{S}$ **then**
17:         Delete edge $(X_i, X_j)$ from $\mathcal{P}$
18:         $sepset(X_i, X_j) \leftarrow \mathbf{S}, sepset(X_j, X_i) \leftarrow \mathbf{S}$
19:     **end if**
20: **end for**
21: **for all** adjacent $(X_i, X_j) \in \mathbf{B}$ **do**
22:     Replace edge $(X_i, X_j)$ with $X_i \circ\!\!-\!\!\circ X_j$ in $\mathcal{P}$
23: **end for**
24: Apply $\mathcal{R}_0 - \mathcal{R}_4, \mathcal{R}_8 - \mathcal{R}_{10}$ to $\mathcal{P}$

---

$\{\mathbf{S}_1, \ldots, \mathbf{S}_m\}$ for $An(\mathbf{Y} \setminus \mathbf{W})_{G_{\mathbf{XW}}}$ and $An(\mathbf{Y} \setminus \mathbf{W})_{H_{\mathbf{XW}}}$. Moreover, $G$ and $H$ have the same c-components $\{\mathbf{C}_1, \ldots, \mathbf{C}_m\}$ where $\mathbf{C}_1 \supseteq \mathbf{S}_1, \mathbf{C}_2 \supseteq \mathbf{S}_2, \ldots$, and $\mathbf{C}_m \supseteq \mathbf{S}_m$. Therefore, $\mathrm{Pr}_{\mathbf{XW}}(\mathbf{Y} \setminus \mathbf{W})$ is identifiable in $G$ iff it is identifiable in $H$. Since all variables in the c-components have the same parents in $G$ and $H$, the formula returned by IDENTIFY applied on $H$ can be used to identify $\mathrm{Pr}_{\mathbf{xw}}(\mathbf{y} \setminus \mathbf{w})$ in $G$ according to (Tian & Pearl, 2003, Lemma 4). $\qquad\square$

### D.2 PROOF OF PROPOSITION 3.3

*Proof.* The proof is based on constructing a causal graph $G$ such that Proposition 3.2 is violated under any such $\mathcal{L}, \mathcal{U}, \mathcal{L}', \mathcal{U}'$. Let $X$ be a treatment variable where $\Gamma(X) = \mathcal{L}$ and $Y$ be an outcome variable where $\Gamma(Y) = \mathcal{U}$. Suppose $\mathcal{L} = \mathcal{U}$, we simply construct $G$ with two edges: $X \rightarrow Y$ and $X \leftrightarrow Y$. The causal effect $\mathrm{Pr}_{\mathbf{X}}(\mathbf{Y})$ is unidentifiable in this case. However, any subgraph obtained under $(\mathcal{L}', \mathcal{U}')$ does not contain $X, Y$; hence, $\mathrm{Pr}_{\mathbf{X}}(\mathbf{Y})$ is identifiable in these subgraphs, which means that the identifiability is not preserved. Suppose now $\mathcal{L} < \mathcal{U}$, let $W$ be a non-treatment variable where $\Gamma(W) = \mathcal{L}$. We construct $G$ with the following edges: $X \rightarrow W, X \leftrightarrow W, W \rightarrow Y$, where $\mathrm{Pr}_{\mathbf{X}}(\mathbf{Y})$ is unidentifiable. However, the causal effect becomes identifiable either when $\mathcal{L}' > \mathcal{L}$, which removes $X, W$ from $G$, or when $\mathcal{U}' < \mathcal{U}$, which removes $Y$ from $G$. $\qquad\square$

### D.3 PROOF OF PROPOSITION 3.4

*Proof.* Consider the following chain graph $V_1 \rightarrow V_2 \rightarrow \cdots \rightarrow V_t$ where $V_i$ belongs to $i^{\text{th}}$ tier and the causal effect $\mathrm{Pr}_{V_{t-1}}(V_t)$. As $t$ grows, the number of edges in the chain becomes unbounded. However, the size of pruned graph remains $V_{t-1} \rightarrow V_t$, which contains a single edge. $\qquad\square$

### D.4 PROOF OF PROPOSITION 3.5

We start by showing that $\mathrm{Pr}_{\mathbf{X}}(\mathbf{Y}|\mathbf{Z})$ is identifiable in $G$ (or $H$) iff $\mathrm{Pr}_{\mathbf{X}}(\mathbf{Y}, \mathbf{Z})$ is identifiable in $G$ (or $H$). Consider the following equality based on do-calculus, Rule 3:

$$\mathrm{Pr}_{\mathbf{X}}(\mathbf{Y}|\mathbf{Z}) = \frac{\mathrm{Pr}_{\mathbf{X}}(\mathbf{Y}, \mathbf{Z})}{\mathrm{Pr}_{\mathbf{X}}(\mathbf{Z})} = \frac{\mathrm{Pr}_{\mathbf{X}}(\mathbf{Y}, \mathbf{Z})}{\mathrm{Pr}(\mathbf{Z})} \qquad (2)$$

Since $\Pr(\mathbf{Z})$ is fixed by the observational distribution, $\Pr_{\mathbf{X}}(\mathbf{Y}|\mathbf{Z})$ is identifiable iff $\Pr_{\mathbf{X}}(\mathbf{Y}, \mathbf{Z})$ is identifiable. Since $\Gamma^+(\mathbf{Z}) < \Gamma^-(\mathbf{X})$, the graph $H$ is identical to the subgraph induced by Proposition 3.2 for $\Pr_{\mathbf{X}}(\mathbf{Y}, \mathbf{Z})$. Hence, $\Pr_{\mathbf{X}}(\mathbf{Y}, \mathbf{Z})$ is identifiable in $G$ iff it is identifiable in $H$, which concludes that $\Pr_{\mathbf{X}}(\mathbf{Y}|\mathbf{Z})$ is identifiable in $G$ iff it is identifiable in $H$. The identifying formula for $\Pr_{\mathbf{X}}(\mathbf{Y}|\mathbf{Z})$ follows from Eq. 2.

### D.5 PROOF OF PROPOSITION D.2

**Proposition D.2.** *Let $\mathcal{L}, \mathcal{U}, \mathcal{L}', \mathcal{U}'$ be positive integers satisfying $\mathcal{L} \leq \mathcal{U}$, $\mathcal{L}' \leq \mathcal{U}'$, and at least one of $\mathcal{L}' > \mathcal{L}$ or $\mathcal{U}' < \mathcal{U}$ holds. Then there exists a causal graph $G$ and a tier mapping $\Gamma$ for which $\Gamma^-(\mathbf{X}) = \mathcal{L}$, $\Gamma^+(\mathbf{Y}) = \mathcal{U}$, and Proposition 3.5 no longer holds if the bounds $\Gamma^-(\mathbf{X})$ and $\Gamma^+(\mathbf{Y})$ are replaced with $\mathcal{L}'$ and $\mathcal{U}'$.*

*Proof.* This follows directly from the proof of Proposition 3.3. $\qquad\square$

### D.6 PROOF OF PROPOSITION 3.6

*Proof.* According to (Shpitser & Pearl, 2008), $\Pr_{\mathbf{X}}(\mathbf{Y}|\mathbf{Z})$ is identifiable in $G$ iff $\Pr_{\mathbf{X}\mathbf{Z}_1}(\mathbf{Y}, \mathbf{Z}_2)$ where $\mathbf{Z}_1$ and $\mathbf{Z}_2$ partition $\mathbf{Z}$. Moreover, $\mathbf{Z}_1$ is the unique maximal subset of $\mathbf{Z}$ such that $\Pr_{\mathbf{X}}(\mathbf{Y}|\mathbf{Z}) = \Pr_{\mathbf{X}\mathbf{Z}_1}(\mathbf{Y}|\mathbf{Z}_2)$ by rule 2 of do-calculus. This set $\mathbf{Z}_1$ is the same in $G$ and $H$ since $(\mathbf{Y} \perp\!\!\!\perp \mathbf{Z}_1 | \mathbf{X}, \mathbf{Z}_2)_{G_{\overline{\mathbf{X}}\underline{\mathbf{Z}_1}}}$ iff $(\mathbf{Y} \perp\!\!\!\perp \mathbf{Z}_1 | \mathbf{X}, \mathbf{Z}_2)_{H_{\overline{\mathbf{X}}\underline{\mathbf{Z}_1}}}$ when $H$ only removes edges below $\mathbf{Y} \cup \mathbf{Z}$. $\qquad\square$

### D.7 PROOF OF PROPOSITION 4.1

*Proof.* By construction, the partial PAG $\mathcal{P}' = \bigcup_{i=minTier}^{maxTier} \mathcal{P}^i$ if we check each case in Propositions 3.2-3.6. Moreover, the soundness and completeness of learning each partial graph $\mathcal{P}^i$ follows directly from (Andrews et al., 2020, Lemma 12). $\qquad\square$

### D.8 PROOF OF PROPOSITION 4.2

*Proof.* WLG, let $\mathbf{A} = \{A_1, \ldots, A_n\}$ be the variables in the first tier, and $\mathbf{B} = \{B_1, \ldots, B_n\}$ be the variables in the second tier. We add bidirected edges between every pair of variables in $\mathbf{A}$, directed edges from each $A_i$ to $B_i$, and directed edges from each $B_i$ to $B_{i+1}$. Suppose we are interested in the causal effect $\Pr_{B_1}(B_n)$. It takes $O(n^2 \cdot 2^n)$ conditional independence (CI) tests for tiered FCI algorithm to learn all the edges among $\mathbf{A}$. In contrast, Algorithm 1 ignores all edges among $\mathbf{A}$ and learns the T-component $G^2$ by conditioning on at most 2 variables. Since there are $O(n^2)$ possible edges and $O(n^2)$ possible conditioning sets, Algorithm 1 takes at most $O(n^4)$ CI tests. $\qquad\square$

| $T$ | $|\mathbf{Z}|$ | Adjacency Precision | Adjacency Recall | Arrowhead Precision | Arrowhead Recall | Time (s) |
|---|---|---|---|---|---|---|
| 5 | Full | 0.96 | 0.97 | 0.86 | 0.94 | 22.13 |
|   | 0 | 0.97 | 0.97 | 0.88 | 0.95 | 11.89 |
|   | 1 | 0.97 | 0.97 | 0.87 | 0.95 | 14.31 |
|   | 2 | 0.97 | 0.97 | 0.86 | 0.95 | 16.53 |
|   | 3 | 0.96 | 0.97 | 0.86 | 0.95 | 17.64 |
| 6 | Full | 0.95 | 0.97 | 0.85 | 0.95 | 32.37 |
|   | 0 | 0.96 | 0.98 | 0.88 | 0.96 | 19.62 |
|   | 1 | 0.96 | 0.97 | 0.86 | 0.95 | 24.57 |
|   | 2 | 0.95 | 0.97 | 0.86 | 0.95 | 26.94 |
|   | 3 | 0.96 | 0.97 | 0.86 | 0.95 | 27.30 |
| 7 | Full | 0.95 | 0.97 | 0.85 | 0.95 | 46.57 |
|   | 0 | 0.95 | 0.98 | 0.86 | 0.95 | 26.50 |
|   | 1 | 0.95 | 0.97 | 0.87 | 0.94 | 31.77 |
|   | 2 | 0.95 | 0.98 | 0.86 | 0.95 | 35.16 |
|   | 3 | 0.95 | 0.97 | 0.86 | 0.95 | 38.33 |
| 8 | Full | 0.94 | 0.97 | 0.84 | 0.94 | 65.83 |
|   | 0 | 0.95 | 0.97 | 0.87 | 0.95 | 38.80 |
|   | 1 | 0.95 | 0.97 | 0.86 | 0.95 | 45.13 |
|   | 2 | 0.94 | 0.97 | 0.86 | 0.95 | 49.63 |
|   | 3 | 0.94 | 0.97 | 0.86 | 0.95 | 50.09 |
| 9 | Full | 0.93 | 0.96 | 0.84 | 0.94 | 89.52 |
|   | 0 | 0.94 | 0.97 | 0.86 | 0.94 | 64.60 |
|   | 1 | 0.93 | 0.97 | 0.85 | 0.94 | 71.43 |
|   | 2 | 0.93 | 0.97 | 0.84 | 0.94 | 74.32 |
|   | 3 | 0.93 | 0.97 | 0.84 | 0.94 | 75.64 |
| 10 | Full | 0.94 | 0.97 | 0.85 | 0.94 | 123.04 |
|   | 0 | 0.94 | 0.97 | 0.86 | 0.94 | 74.69 |
|   | 1 | 0.94 | 0.97 | 0.85 | 0.94 | 87.89 |
|   | 2 | 0.94 | 0.97 | 0.85 | 0.94 | 92.42 |
|   | 3 | 0.94 | 0.97 | 0.85 | 0.94 | 99.82 |

Table 1: Adjacency Precision/Recall, Arrowhead Precision/Recall, and time for FCITIERS (rows with $|\mathbf{Z}|=Full$) and Algorithm 1 (rows with $|\mathbf{Z}| \in \{0, 1, 2, 3\}$) under varying numbers of tiers ($T$).

