# OpenReview forum: "Query-Specific Causal Graph Pruning Under Tiered Knowledge"
_ICLR.cc/2026/Conference — ICLR 2026 Poster_

### Official Review · Reviewer_hPcC · 2025-10-24

**Soundness:** 3
**Presentation:** 3
**Contribution:** 3
**Rating:** 6
**Confidence:** 3

**Summary:**

The paper focuses on the challenge of answering causal queries in complex systems without having to fully specify or learn an entire causal graph. This comes from the motivation that in some settings the causal model can involve many variables, but a researcher may only be interested in the effect of certain treatments on certain outcomes. The paper focuses on how incorporating background knowledge (tiered knowledge) about the system can improve both identifiability of causal effects and computational efficiency.   They introduce a graph pruning approach to remove certain edges from the causal graph without affecting the identifiability (causal identification on simplified graphs). Based on this, the paper introduces a query-specific causal discovery algorithm that takes as input not just observational data but also a specific causal query of interest to only learn the portion of the causal graph relevant to that query. The paper shows theoretically and empirically that their method can achieve exponential speedups

**Strengths:**

- I think the paper is clear and well written. Sometimes a bit too heavy in notation, but this is perhaps necessary for introducing the theoretical results. I would encourage having more intuitive introductions to some of the notations (for instance c-components/tiered knowledge). I enjoyed the graph examples in the paper, they made the understanding easier.
- the research question is relevant in the field, further improvements in this direction can potentially help causal effect estimation be more applicable in practical scenarios. It is also important to start from this kind of theoretically-grounded contribution to understand to what extent identifiability can hold with different kinds of domain knowledge.
- the theoretical results show that the proposed algorithm is sound and complete
- the experimental result show promising results in terms of efficiency without sacrificing correctness of the learned structures

**Weaknesses:**

- I think the paper could be strengthened by adding some examples of real world applications in introduction where tier knowledge is available to practitioners
- The authors should explicitly mention as well that this paper relies on acyclicity of the causal structure
- in Line 078, the notation $Pr_{B_1}(B_3,C_3)$ is introduced, however the subscript notation had not previously been introduced. The authors should fix this.
- related to the previous sentence, also when mentioning 'they are identifiable in the pruned graph G' I would add a reference to Fig 1b, to make it easier for the reader to understand the example.
- Minor: format of the title seems different from the ICLR layout

**Questions:**

- can you provide some examples of practical applications where we actually have tiered knowledge?
- It is hard to imagine a practical setting in which we can rule out bidirected edges across tiers. I know that the authors mention that this extension is left for future work, but what kind of steps would be necessary for this extension?
- it seems that the advantages of the algorithm decrease be increasing the size of the conditioning set $Z$. What happens when this is substantially larger? It would also be nice to see the improvement in terms of num tiers as percentage of the entire set of nodes, and then analyse how this changes with different sizes of underlying causal graph

---

> ### Author Response · Authors · 2025-11-25
>
> > I would encourage having more intuitive introductions to some of the notations (for instance c-components/tiered knowledge)
>
> Thanks for your suggestion. We will incorporate this.
>
> > I think the paper could be strengthened by adding some examples of real world applications in introduction where tier knowledge is available to practitioners
> > can you provide some examples of practical applications where we actually have tiered knowledge?
>
> We will incorporate additional real-world applications of tiered knowledge (the first two were mentioned in [1], and the third is new):
>
> 1.  **Time-series features with contemporaneous confounding.** This arises when hidden confounders only influence variables within the same timestamp. For example, consider a classroom scenario with four variables at each day $t$: lecture difficulty ($D_t$), the student’s learning effort ($E_t$), daily Quiz score ($Q_t$), and daily homework score ($H_t$). Suppose variables $E_t$, $Q_t$, and $H_t$ are observed and are confounded by the hidden variable $D_t$, then the variables {$E_1, Q_1, H_1$}, {$E_2, Q_2, H_2$}, … {$E_t, Q_t, H_t$} exhibit tiered knowledge, since no hidden confounder spans across different timestamps. However, there can be cross-tier directed edges, such as $Q_{t} \rightarrow E_{t+1}$, $E_{t} \rightarrow E_{t+1}$, capturing temporal dependencies.
>
> 2. **Context variables** Consider a medical example with _context variables_ Month ($M$), Region ($R$), Temperature ($T$), and _system variables_ Sick ($S$), Recovery ($R$) and Immunity ($I$) for some patient. Suppose all variables except $I$ are observed. We may add edges such as $M \rightarrow T$, $R \rightarrow T$, $T \rightarrow S$, $I \rightarrow S$, $I \rightarrow R$, $S \rightarrow R$. It is clear that all context variables belong to the first tier and all system variables belong to the second tier, since context variables can cause system variables but not vice versa, and there is no hidden confounder between the two tiers. This setting has been studied extensively in the joint causal inference (JCI) framework [3], which also accommodates different interventions.
>
> 3. **Hierarchical modeling.** Tiered knowledge naturally appears when variables can be grouped into hierarchies, with confounding restricted to occur only within each hierarchy (tier). Consider the following education system example with three tiers. The _district-tier_ contains observed variables “District Education Funding (DF)” and “Population (DP)”. The _school-tier_ contains observed variables “Teaching Quality (ST)” and “Extracurricular Activities (SE)”, and a hidden confounder “School Funding (SF)”. The _student-tier_ contains observed variables “Test Grades (OT)” and “College Admission (OC)”, and a hidden confounder “Motivation (OM)”. We can then construct a causal graph with tiered knowledge by adding edges $DF \rightarrow SF$, $DP \rightarrow SF$, $SF \rightarrow ST$, $SF \rightarrow SE$,  $ST \rightarrow OT$, $OT \rightarrow OC$, $SE \rightarrow OC$, $OM \rightarrow OT$, $OM \rightarrow OC$.
>
>
> > It is hard to imagine a practical setting in which we can rule out bidirected edges across tiers. I know that the authors mention that this extension is left for future work, but what kind of steps would be necessary for this extension?
>
> Great question. We have not explored the case with cross-tier bidirected edges yet. However, the results in this work should still be applicable if the tiers (without cross-tier bidirected edges) can be learned from data. This would likely require new methods for identifying confounders that span multiple tiers.
>
> > it seems that the advantages of the algorithm decrease be increasing the size of the conditioning set Z. What happens when this is substantially larger?
>
> It depends on the location of Z. For example, in the worst case, if the variables in Z span all tiers, then no pruning can be performed by Algorithm 1 (line 5). In contrast, if Z contains only variables from higher tiers, the preprocessing step (Algorithm 1, lines 1–6) remains effective in pruning tiers and can result in significant speedups.

---

> ### Author Response · Authors · 2025-11-25
>
> > It would also be nice to see the improvement in terms of num tiers as percentage of the entire set of nodes, and then analyse how this changes with different sizes of underlying causal graph
>
> Thanks for the suggestion. We have rerun the experiments and reported the number of tiers in the pruned graph relative to the full graph. The results are shown below (%tiers_learned indicates the relative number of tiers in the pruned graph):
>
> | $T$ | $\lvert\textbf{Z}\rvert$ | %tiers_learned | discovery_time_full (s) | discovery_time_partial (s) | ID_time_full (s) | ID_time_partial (s) |
> |---|---|---|---|---|---|---|
> | 5 | 0 | 66% | 39.3 | 27.2 | 0.009 | 0.008 |
> | 5 | 1 | 78% | 39.3 | 30.0 | 0.014 | 0.012 |
> | 5 | 2 | 85% | 39.3 | 31.6 | 0.019 | 0.018 |
> | 5 | 3 | 89% | 39.3 | 33.7 | 0.021 | 0.020 |
> | 6 | 0 | 75% | 67.1 | 51.1 | 0.010 | 0.008 |
> | 6 | 1 | 83% | 67.1 | 52.6 | 0.012 | 0.009 |
> | 6 | 2 | 93% | 67.1 | 59.2 | 0.015 | 0.014 |
> | 6 | 3 | 95% | 67.1 | 60.4 | 0.019 | 0.011 |
> | 7 | 0 | 60% | 86.8 | 53.1 | 0.009 | 0.006 |
> | 7 | 1 | 81% | 86.8 | 62.3 | 0.014 | 0.011 |
> | 7 | 2 | 85% | 86.8 | 66.9 | 0.015 | 0.013 |
> | 7 | 3 | 90% | 86.8 | 70.8 | 0.019 | 0.017 |
> | 8 | 0 | 63% | 137.4 | 82.6 | 0.017 | 0.012 |
> | 8 | 1 | 74% | 137.4 | 90.5 | 0.021 | 0.017 |
> | 8 | 2 | 78% | 137.4 | 93.3 | 0.025 | 0.021 |
> | 8 | 3 | 82% | 137.4 | 97.8 | 0.032 | 0.028 |
> | 9 | 0 | 50% | 178.5 | 86.7 | 0.012 | 0.007 |
> | 9 | 1 | 70% | 178.5 | 114.3 | 0.014 | 0.010 |
> | 9 | 2 | 83% | 178.5 | 130.5 | 0.020 | 0.016 |
> | 9 | 3 | 86% | 178.5 | 141.3 | 0.021 | 0.020 |
> | 10 | 0 | 60% | 257.9 | 174.9 | 0.017 | 0.012 |
> | 10 | 1 | 78% | 257.9 | 199.5 | 0.020 | 0.017 |
> | 10 | 2 | 84% | 257.9 | 213.1 | 0.025 | 0.021 |
> | 10 | 3 | 92% | 257.9 | 214.0 | 0.035 | 0.032 |
>
>
> > The authors should explicitly mention as well that this paper relies on acyclicity of the causal structure
>
> > in Line 078, the notation is introduced, however the subscript notation had not previously been introduced. The authors should fix this.
>
> >related to the previous sentence, also when mentioning 'they are identifiable in the pruned graph G' I would add a reference to Fig 1b, to make it easier for the reader to understand the example.
>
> > Minor: format of the title seems different from the ICLR layout
>
> Will fix these. Thanks.
>
> References:
>
> [1] Bryan Andrews, Peter Spirtes, and Gregory F. Cooper. On the completeness of causal discovery in the presence of latent confounding with tiered background knowledge. AISTATS, 2020.
>
> [2] Joris M. Mooij, Sara Magliacane, and Tom Claassen. Joint Causal Inference from Multiple Contexts. J. Mach. Learn. Res. 2020.

---

> > ### Comment · Reviewer_hPcC · 2025-11-27
> > **Thank you**
> >
> > I thank the authors for the reply. I will keep my score as it is.

---

### Official Review · Reviewer_xo6S · 2025-10-27

**Soundness:** 3
**Presentation:** 3
**Contribution:** 3
**Rating:** 6
**Confidence:** 2

**Summary:**

This paper studies the identification and computation formula for causal (treament) effect and conditional causal effect under tiered knowledge. It shows the identification and computation can be infered from a simplified (pruned) graph based on the original graph using the tiered knowledge. The pruning procedure is given and a query-specific causal discovery method is proposed based on the pruning procedure.

**Strengths:**

- The paper well-motivated. It introduces the concept of query-specific learning, which reframes causal discovery from global reconstruction to task-focused learning, suitable when full-graph recovery is unnecessary or infeasible.
- The exposition is generally clear and the examples (especially motivating ones in the introduction) make the idea intuitive.

**Weaknesses:**

See questions.

**Questions:**

- One of the main message of this work is: to check the idenfication of a given (conditional) causal effect, it suffices to look at a prune graph, then use the existing identification formula. However, it there truely benefit of doing so in terms of computation? Would calling identification formula on the whole graph reduce to calling identification formula on the pruned graph? What is the gain of using the pruned graph when there is an extra step to prune the graph?
- The experiment only shows the speedup against existing methods on causal discovery. With the main motivation to be causal effect identification check, would there be experiment showing the speedup in this check?
- Why does the experiment in Figure 5 seem not depend on $|Z|$?

---

> ### Author Response · Authors · 2025-11-25
>
> >One of the main message of this work is: to check the idenfication of a given (conditional) causal effect, it suffices to look at a prune graph, then use the existing identification formula. However, it there truely benefit of doing so in terms of computation? Would calling identification formula on the whole graph reduce to calling identification formula on the pruned graph? What is the gain of using the pruned graph when there is an extra step to prune the graph?
>
> This is a great question. The main benefit of the pruning results lies in the causal graph specification. A pruned graph contains fewer edges, so one can specify (or learn) a smaller graph and use it for causal identification. When the causal graph is specified manually from domain knowledge, this means that only the edges in the pruned graph need to be marked (i.e., known) to compute causal effects. When the graph is learned from data, we showed in Section 4 that the causal discovery process can be accelerated significantly because a smaller graph needs to be learned. We will emphasize this further.
>
> The identification formula for a pruned graph may differ from that of the full graph. The valid identification formulas for pruned graphs are provided in the propositions (different formulas for pruned and full graphs in Propositions 3.2 and 3.5, and the same formula in Proposition 3.6). See also the example in lines 276–284, which illustrates that not all identification formulas for pruned graphs remain valid for the full graph.
>
> > The experiment only shows the speedup against existing methods on causal discovery. With the main motivation to be causal effect identification check, would there be experiment showing the speedup in this check?
>
> Causal-effect identification checks (e.g., Identify, ID, IDC) run in polynomial time and are much faster than causal discovery. Hence, the time for identification checks becomes negligible in practice. Following the reviewer’s suggestion, we conducted additional experiments to include the time for identification checks, and the results are shown below. ID_time_full reports the execution time for identification checks on the full graphs, and ID_time_partial reports the time on the partial (pruned) graphs.
>
> | $T$ | $\lvert\textbf{Z}\rvert$ | %tiers_learned | discovery_time_full (s) | discovery_time_partial (s) | ID_time_full (s) | ID_time_partial (s) |
> |---|---|---|---|---|---|---|
> | 5 | 0 | 66% | 39.3 | 27.2 | 0.009 | 0.008 |
> | 5 | 1 | 78% | 39.3 | 30.0 | 0.014 | 0.012 |
> | 5 | 2 | 85% | 39.3 | 31.6 | 0.019 | 0.018 |
> | 5 | 3 | 89% | 39.3 | 33.7 | 0.021 | 0.020 |
> | 6 | 0 | 75% | 67.1 | 51.1 | 0.010 | 0.008 |
> | 6 | 1 | 83% | 67.1 | 52.6 | 0.012 | 0.009 |
> | 6 | 2 | 93% | 67.1 | 59.2 | 0.015 | 0.014 |
> | 6 | 3 | 95% | 67.1 | 60.4 | 0.019 | 0.011 |
> | 7 | 0 | 60% | 86.8 | 53.1 | 0.009 | 0.006 |
> | 7 | 1 | 81% | 86.8 | 62.3 | 0.014 | 0.011 |
> | 7 | 2 | 85% | 86.8 | 66.9 | 0.015 | 0.013 |
> | 7 | 3 | 90% | 86.8 | 70.8 | 0.019 | 0.017 |
> | 8 | 0 | 63% | 137.4 | 82.6 | 0.017 | 0.012 |
> | 8 | 1 | 74% | 137.4 | 90.5 | 0.021 | 0.017 |
> | 8 | 2 | 78% | 137.4 | 93.3 | 0.025 | 0.021 |
> | 8 | 3 | 82% | 137.4 | 97.8 | 0.032 | 0.028 |
> | 9 | 0 | 50% | 178.5 | 86.7 | 0.012 | 0.007 |
> | 9 | 1 | 70% | 178.5 | 114.3 | 0.014 | 0.010 |
> | 9 | 2 | 83% | 178.5 | 130.5 | 0.020 | 0.016 |
> | 9 | 3 | 86% | 178.5 | 141.3 | 0.021 | 0.020 |
> | 10 | 0 | 60% | 257.9 | 174.9 | 0.017 | 0.012 |
> | 10 | 1 | 78% | 257.9 | 199.5 | 0.020 | 0.017 |
> | 10 | 2 | 84% | 257.9 | 213.1 | 0.025 | 0.021 |
> | 10 | 3 | 92% | 257.9 | 214.0 | 0.035 | 0.032 |
>
> > Why does the experiment in Figure 5 seem not depend on |Z|?
>
> The experimental results do depend on |Z|. We encourage the reviewer to check out Table 1 in the Appendix (page 15), which demonstrates this dependence more clearly than Figure 5.

---

### Official Review · Reviewer_z3UD · 2025-10-31

**Soundness:** 1
**Presentation:** 2
**Contribution:** 1
**Rating:** 2
**Confidence:** 3

**Summary:**

A method is presented that prunes edges from causal graphs under tiered knowledge. Conditions for the removal of edges are specified that don't affect the identifiability of (conditional) causal effects. These are subsequently used to develop a query-specific causal discovery procedure.

**Strengths:**

The presented method is potentially computationally more efficient than the FCITiers algorithm it is based on. The presentation is clear.

**Weaknesses:**

- Bidirected edges across tiers are not allowed, which is a strong limitation. In fact, in the medical context example (lines 121-123) used to justify the assumption of tiers, I don't believe it can be assumed that there are no latent confounders between, say, treatment and recovery. This restriction makes me doubt whether this paper is appropriate for a top conference like ICLR.
- The paper conflates ADMGs and MAGs. Initially, ADMGs are used for causal graphs, and their intuitive semantics are explained. But the theoretical developments later in the paper use MAGs and PAGs, which have different semantics. I am not sure the presented results are correct if the ground-truth graph is an ADMG that is not a MAG.
- The pruning method seems to be incremental compared to existing pruning techniques.
- Proposition 4.2 proves the possibility of exponential speedup, but this only applies to "a class of distributions". This makes the statement rather weak. (And indeed, the proof constructs a family of graphs that contains one graph for each $n$.)

**Questions:**

- How is the data generated in the experiments?
- How are propositions 3.5 and 3.6 related? Would it be possible to just use Proposition 3.6?
- How does this work compare to existing work using pruning, such as [1]?

[1] Tikka, Santtu, and Juha Karvanen. "Enhancing identification of causal effects by pruning." Journal of Machine Learning Research 18.194 (2018): 1-23.

### Other comments
- "casual" -> "causal" in line 97

---

> ### Author Response · Authors · 2025-11-25
>
> > Bidirected edges across tiers are not allowed, which is a strong limitation. In fact, in the medical context example (lines 121-123) used to justify the assumption of tiers, I don't believe it can be assumed that there are no latent confounders between, say, treatment and recovery. This restriction makes me doubt whether this paper is appropriate for a top conference like ICLR.
>
>
> Tiered knowledge (without bidirected edges across tiers) was introduced in [2], and we adopted the same definition here (see lines 69-73, 113-116). In lines 71-73, we mentioned several applications of tiered knowledge and referred readers to [2] for further discussion. That said, we will incorporate additional real-world applications of tiered knowledge and replace lines 121-123 with the following (the first two were mentioned in [2], and the third is new):
>
> 1.  **Time-series features with contemporaneous confounding.** This arises when hidden confounders only influence variables within the same timestamp. For example, consider a classroom scenario with four variables at each day $t$: lecture difficulty ($D_t$), the student’s learning effort ($E_t$), daily Quiz score ($Q_t$), and daily homework score ($H_t$). Suppose variables $E_t$, $Q_t$, and $H_t$ are observed and are confounded by the hidden variable $D_t$, then the variables {$E_1, Q_1, H_1$}, {$E_2, Q_2, H_2$}, … {$E_t, Q_t, H_t$} exhibit tiered knowledge, since no hidden confounder spans across different timestamps. However, there can be cross-tier directed edges, such as $Q_{t} \rightarrow E_{t+1}$, $E_{t} \rightarrow E_{t+1}$, capturing temporal dependencies.
>
> 2. **Context variables** Consider a medical example with _context variables_ Month ($M$), Region ($R$), Temperature ($T$), and _system variables_ Sick ($S$), Recovery ($R$) and Immunity ($I$) for some patient. Suppose all variables except $I$ are observed. We may add edges such as $M \rightarrow T$, $R \rightarrow T$, $T \rightarrow S$, $I \rightarrow S$, $I \rightarrow R$, $S \rightarrow R$. It is clear that all context variables belong to the first tier and all system variables belong to the second tier, since context variables can cause system variables but not vice versa, and there is no hidden confounder between the two tiers. This setting has been studied extensively in the joint causal inference (JCI) framework [3], which also accommodates different interventions.
>
> 3. **Hierarchical modeling.** Tiered knowledge naturally appears when variables can be grouped into hierarchies, with confounding restricted to occur only within each hierarchy (tier). Consider the following education system example with three tiers. The _district-tier_ contains observed variables “District Education Funding (DF)” and “Population (DP)”. The _school-tier_ contains observed variables “Teaching Quality (ST)” and “Extracurricular Activities (SE)”, and a hidden confounder “School Funding (SF)”. The _student-tier_ contains observed variables “Test Grades (OT)” and “College Admission (OC)”, and a hidden confounder “Motivation (OM)”. We can then construct a causal graph with tiered knowledge by adding edges $DF \rightarrow SF$, $DP \rightarrow SF$, $SF \rightarrow ST$, $SF \rightarrow SE$,  $ST \rightarrow OT$, $OT \rightarrow OC$, $SE \rightarrow OC$, $OM \rightarrow OT$, $OM \rightarrow OC$.
>
> > The paper conflates ADMGs and MAGs. Initially, ADMGs are used for causal graphs, and their intuitive semantics are explained. But the theoretical developments later in the paper use MAGs and PAGs, which have different semantics. I am not sure the presented results are correct if the ground-truth graph is an ADMG that is not a MAG.
>
> We assume that the underlying causal graphs are MAGs when considering the causal discovery method in Section 4. This assumption is standard when learning causal structure from data, as in FCI [4], GFCI [5], and FCI-JCI [3]. Since MAGs are a subclass of ADMGs, all pruning results remain valid when the ground truth is a MAG, thereby justifying the correctness of our causal discovery algorithm. We will clarify this further at the beginning of Section 4.
>
> References:
>
> [1] Tikka, Santtu, and Juha Karvanen. "Enhancing identification of causal effects by pruning." Journal of Machine Learning Research 18.194 (2018): 1-23.
>
> [2] Bryan Andrews, Peter Spirtes, and Gregory F. Cooper. On the completeness of causal discovery in the presence of latent confounding with tiered background knowledge. AISTATS, 2020.
>
> [3] Joris M. Mooij, Sara Magliacane, and Tom Claassen. Joint Causal Inference from Multiple Contexts. J. Mach. Learn. Res. 2020.
>
> [4] Peter Spirtes, Clark Glymour, and Richard Scheines. Causation, Prediction, and Search, Second Edition. Adaptive computation and machine learning. MIT Press, 2000.
>
> [5] J. M. Ogarrio, P. Spirtes, J. Ramsey, A hybrid causal search algorithm for latent
> variable models, in: Conference on Probabilistic Graphical Models, PMLR, 2016,
> pp. 368–379.

---

> ### Author Response · Authors · 2025-11-25
>
> > The pruning method seems to be incremental compared to existing pruning techniques.
>
> To the best of our knowledge, no prior study has investigated graph pruning using tiered knowledge. Please let us know if you are aware of any relevant work.
>
> > Proposition 4.2 proves the possibility of exponential speedup, but this only applies to "a class of distributions". This makes the statement rather weak. (And indeed, the proof constructs a family of graphs that contains one graph for each n.
>
> Like many complexity-theoretic arguments, Proposition 4.2 is intended as a possibility result: it shows that exponential speedup exists for a family of graphs, not that it holds for all distributions. Establishing universal exponential speedups for _all distributions/graphs_ is impossible here since the speedups in Propositions 3.2-3.6 rely on the tiered knowledge that is itself determined by the graph structure – for example, no pruning is possible if all variables belong to the same tier. We will clarify this further.
>
> > How is the data generated in the experiments?
>
> Good catch. The data are sampled from causal graphs with randomly generated parameterizations (i.e., randomly generated conditional probability distributions). We will add this to the experimental setup.
>
> > How are propositions 3.5 and 3.6 related? Would it be possible to just use Proposition 3.6?
>
> Propositions 3.5 and 3.6 are fundamentally different as we emphasized in lines 360-369, 377-392. Specifically, Proposition 3.5 can only be applied when the condition on line 320 holds, whereas Proposition 3.6 can be applied in all scenarios. Hence, using only Proposition 3.6 would miss opportunities for pruning that Proposition 3.5 enables.
>
> > How does this work compare to existing work using pruning, such as [1]?
>
> The pruning methods in this work are fundamentally different from the ones in [1]. Methods in [1] prune _variables_ from causal graphs through latent projection, while the results (Propositions 3.2-3.6) in our work prune _edges_ from causal graphs and can _only be applied when tiered knowledge is available_. We will discuss these differences at the beginning of Section 3.1.

---

### Author Response · Authors · 2025-11-25

We thank the reviewers for their thoughtful comments and suggestions. Please see individual responses below.

---

### Meta-Review · Area_Chair_wN5n · 2025-12-22

**Summary:**

According to the reviewers, the main strengths relate to paper clarity (though Reviewer hPcC comments on the heavy notation), quality of writing and addressing a problem that is reasonbly well identified and motivated  in terms of efficiency gain. These aspects have been recognized by all reviewers.

Disputed aspects pertain novelty, which is questioned by Reviewer z3UD that considers the contribution incremental, and theoretical soundness, on which z3UD and hPcC have different views. The former, in particular, finds that the conditions under which Propositions have been derived are not clearly stated in the paper.

A central weakness raised independently by reviewers z3UD and hPcC, instead, concerns the generalizability of the results and the strenght of the assumption concerning the absencre of bidirectional edges between tiers. Both reviewers consider this as an  unrealistic assumption.
Reviewer z3UD viewed this as a major concern, arguing that the experimental evidence is too narrow to support the broader claims.
Reviewer hPcC raised a similar point but considered it a more minor issue, suggesting that additional experiments could potentially address it.

Reviewer xo6S was the least confident reviewer, and parts of their review appear to stem from misunderstandings of the some parts of the work. I am therefore inclided to weigh this review less in the final decision.

**Reviewer Concerns:**

The Authors' rebuttal was covering all the points raised, sometimes in a very concise manner.

The response to the main criticism about the strength of bidirectionality assumption is robustly rooted in early literature, but does not really answer to the point raised about the realism of such an assumption.

About Reviewer z3UD, the response about the issues regarding the conflation of ADMGs and MAGs, the unclarity about the application of propositions 3.5 and 3.6, and the applicability of the exponential speedups are there but do not strenghten the claim of the work (for the former and the third they essentially confirm the limitation). The response about the relationship with earlier works has been partially avoiding the core point of highlighting more clearly the novelty aspects with respect to a broader literature than the, admittingly, brief one referenced in the paper.

Reviewer hPcC explicitly stated they kept the current score post-rebuttal: seemingly the rebuttal satisfied part of the issues raised, but not the entirety (in particular the one about the realism of the bidirectional assumption).

**Reviewer Scores:**

Overall, this is a borderline submission that would have benefited from a deeper and more engaged discussion phase.

There is evidence of a lack of strong reviewer engagement, both in the positive and negative ones, which could have seemingly led to the final average score remaining under 5.5.


Note by the SAC: This meta-review was updated by the SAC.

---

### Decision · Program_Chairs · 2026-01-26

Accept (Poster)